# HɪT-JEPA: A Hɪᴇʀᴀʀᴄʜɪᴄᴀʟ Sᴇʟꜰ-sᴜᴘᴇʀᴠɪsᴇᴅ Tʀᴀ-ᴊᴇᴄᴛᴏʀʏ Eᴍʙᴇᴅᴅɪɴɢ Fʀᴀᴍᴇᴡᴏʀᴋ ꜰᴏʀ Sɪᴍɪʟᴀʀɪᴛʏ Cᴏᴍᴘᴜᴛᴀᴛɪᴏɴ

## Aʙsᴛʀᴀᴄᴛ

The representation of urban trajectory data plays a critical role in effectively ana-
lyzing spatial movement patterns. Despite considerable progress, the challenge of
designing trajectory representations that can capture diverse and complementary
information remains an open research problem. Existing methods struggle in incor-
porating trajectory fine-grained details and high-level summary in a single model,
limiting their ability to attend to both long-term dependencies while preserving
local nuances. To address this, we propose HiT-JEPA (**H**ierarchical **I**nteractions
of **T**rajectory Semantics via a **J**oint **E**mbedding **P**redictive **A**rchitecture), a uni-
fied framework for learning multi-scale urban trajectory representations across
semantic abstraction levels. HiT-JEPA adopts a three-layer hierarchy that progres-
sively captures point-level fine-grained details, intermediate patterns, and high-level
trajectory abstractions, enabling the model to integrate both local dynamics and
global semantics in one coherent structure. Extensive experiments on multiple
real-world datasets for trajectory similarity computation show that HiT-JEPA's
hierarchical design yields richer, multi-scale representations. Code is available at:
`https://anonymous.4open.science/r/HiT-JEPA`.

## 1 Iɴᴛʀᴏᴅᴜᴄᴛɪᴏɴ

With the widespread use of location-aware devices, trajectory data is now produced at an unprece-
dented rate Zhu et al. (2024); Qian et al. (2024). Effectively representing trajectory data powers
critical applications ranging from urban computing applications, such as travel time estimation Chen
et al. (2022b; 2021); Lin et al. (2023), trajectory clustering Fang et al. (2021); Yao et al. (2024); Bai
et al. (2020), and traffic analysis Yu et al. (2017). Trajectories exhibit multi-scale attributes, ranging
from short-term local transitions (e.g., turns and stops) to long-term strategic pathways or routines,
whereas capturing both the fine-grained point-level details of individual trajectories and higher-level
semantic patterns of mobility behavior within a unified framework is challenging. This necessitates a
representation learning model that accommodates this complexity.

Early trajectory analysis methods (heuristic methods) Alt & Godau (1995); Chen & Ng (2004); Chen
et al. (2005); Yi et al. (1998) relied on handcrafted similarity measures and point-matching heuristics.
Recently, deep-learning-based approaches have been applied to learn low-dimensional trajectory
embeddings, alleviating the need for manual feature engineering Yang et al. (2024); Yao et al. (2019);
Yang et al. (2021). Self-supervised learning frameworks Li et al. (2018); Cao et al. (2021), especially
contrastive learning (as shown in Fig. 1, left), further advanced trajectory representation learning by
leveraging large unlabeled datasets Chang et al. (2023); Liu et al. (2022); Li et al. (2024a). However,
these deep learning models usually generate a single scale embedding of an entire trajectory and can-
not integrate different semantic levels, i.e., they often neglect fine-grained point-level information in
favor of broader trajectory-level features. On the other hand, most representation frameworks Chang
et al. (2023); Li et al. (2018) are restricted to a single form of trajectory data encoding and lack a mech-
anism to incorporate global context or higher-level information. Recent work Li et al. (2024b) (as
shown in Fig. 1, middle) explores alternative self-supervised paradigms that capture higher-level se-
mantic information without manual augmentation. Nevertheless, a flexible and semantically aware rep-
resentation architecture that unifies multiple levels of trajectory information remains an open question.

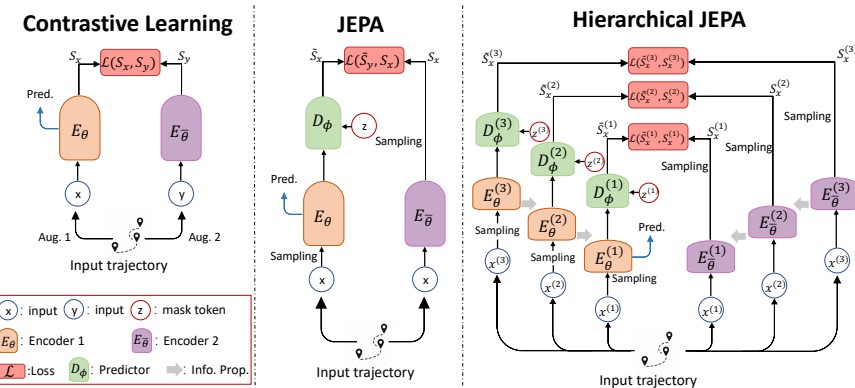

Figure 1: Structural comparisons among Contrastive Learning, JEPA, and Hierarchical JEPA.

Sequence models Vaswani et al. (2017); Hochreiter & Schmidhuber (1997), such as recurrent neural networks (RNNs) and Transformers, are a natural choice for trajectory representation due to their ability to process temporally ordered data. However, they exhibit inherent limitations when representing hierarchical semantics of trajectory data. Specifically, these models often operate at a single temporal granularity: they either overemphasize point-level nuances, making them susceptible to noise, or focus too heavily on coarse trajectory-level summaries and thus oversimplify critical details. This single-scale bias in sequential models prevents them from integrating complementary information across abstraction levels and inhibits explicit semantic interactions between local (point-level), intermediate (segment-level), and global (trajectory-level) representations, making it challenging for sequence models to capture long-term dependencies while maintaining the detailed local nuances. Besides, different from uniformly sampled time series data without spatial topology, trajectories are more capricious due to their irregular, geometry-aware, and network-constrained characteristics.

A new framework is thus required to facilitate the model's understanding of various levels of trajectory representation information, to allow predictions to be grounded on more extensive, multi-dimensional knowledge. In this paper, we propose HiT-JEPA (as shown in Fig. 1, right), a hierarchical framework for urban trajectory representation learning, which is designed to address the gaps mentioned above by integrating trajectory semantics across three levels of granularity. Its three-layer architecture that explicitly captures (1) point-level details, modeling fine-grained spatial-temporal features of consecutive points; (2) intermediate-level patterns, learning representations of local displacement patterns that reflect mesoscopic movement structures; and (3) high-level abstractions, distilling the overall semantic context as summarized moving behaviors of an entire trajectory. The model unifies multiple information scales within a single representation framework through this hierarchy. Moreover, HiT-JEPA enables interactions between adjacent levels to enrich and align the learned trajectory embeddings across scales. By leveraging a joint embedding predictive architecture, the framework learns to predict and align latent representations between these semantic levels, facilitating semantic integration in a self-supervised manner. For clarity, we summarize our contributions as follows:

- We propose HiT-JEPA, a novel hierarchical trajectory representation learning architecture that encapsulates movement information across different semantic levels inside a cohesive framework. HiT-JEPA is the first architecture to explicitly unify both fine-grained and abstract trajectory patterns within a single model.

- HiT-JEPA introduces a joint embedding predictive architecture that unifies the entire trajectory across multiple levels of abstraction, resulting in a flexible representation that can seamlessly incorporate local trajectory nuances and global semantic context. By striking a balance between coarse-to-fine trajectory representations by our proposed hierarchical interaction module, we address the limitations of single-scale or single-view models.

- We conduct extensive experiments on real-world urban trajectory datasets spanning diverse cities and movement patterns, demonstrating that HiT-JEPA's semantically enriched, hierarchical embeddings exhibit comparative trajectory similarity search and remarkably superior zero-shot performance across heterogeneous urban and maritime datasets.

## 2 RELATED WORK

**Urban Trajectory Representation Learning on Similarity Computation.** Self-supervised learning methods for trajectory similarity computation are proposed to cope with robust and generalizable trajectory representation learning on large, unlabeled datasets. t2vec Li et al. (2018) divides spatial regions into rectangular grids and applies Skip-gram Mikolov et al. (2013) models to convert grid cells into word tokens, then leverages an encoder-decoder framework to learn trajectory representations. TrajCL Chang et al. (2023) applies contrastive learning on multiple augmentation schemes with a dual-feature attention module to learn both structural and spatial information in trajectories. CLEAR Li et al. (2024a) proposes a ranked multi-positive contrastive learning method by ordering the similarities of positive trajectories to the anchor trajectories. Recently, T-JEPA Li et al. (2024b) employs a Joint Embedding Predictive Architecture that shifts learning from trajectory data into representation space, establishing a novel self-supervised paradigm for trajectory representation learning. It is also worth noting that robust trajectory representations are often the prerequisite for effective trajectory clustering Yao et al. (2017); Wang et al. (2022); Fang et al. (2021), which focuses on uncovering latent behavioral patterns by grouping trajectories with high semantic affinity. However, none of the above methods manages to explicitly capture hierarchical trajectory information. We propose HiT-JEPA to support coarse-to-fine, multi-scale trajectory abstraction extraction in a hierarchical JEPA structure.

**Hierarchical Self-supervised Learning (HSSL).** Self-supervised learning methods have significantly advanced the capability to extract knowledge from massive amounts of unlabeled data. Recent approaches emphasize multi-scale feature extraction to achieve a more comprehensive understanding of complex data samples (e.g., lengthy texts or high-resolution images with intricate details). In Computer Vision (CV), Chen *et al.* Chen et al. (2022a) stack three Vision Transformers Dosovitskiy et al. (2020) variants (varying patch size configurations) to learn cell, patch, and region representations of gigapixel whole-slide images in computational pathology. Kong *et al.* Kong et al. (2023) design a hierarchical latent variable model incorporating Masked Autoencoders (MAE) He et al. (2022) to encode and reconstruct multi-level image semantics. Xiao *et al.* Xiao et al. (2022) split the hierarchical structure by video semantic levels and employ different learning objectives to capture distinct semantic granularities. In Natural Language Processing (NLP), Zhang *et al.* Zhang et al. (2019) develop HIBERT, leveraging BERT Devlin et al. (2019) to learn sentence-level and document-level text representations for document summarization. Li *et al.* Li et al. (2022) introduce HiCLRE, a hierarchical contrastive learning framework for distantly supervised relation extraction, utilizing Multi-Granularity Recontextualization for cross-level representation interactions to effectively reduce the influence of noisy data. In contrast to these methods, which partition inputs into discrete fragments and directly propagate representations across levels, HiT-JEPA encodes the entire trajectory at multiple abstraction levels by coupling adjacent-level attention weights from a hierarchical JEPA to learn multi-scale urban trajectory representations.

## 3 METHODOLOGY

Compared to previous methods that only model trajectories at point-level, our primary goal in designing HiT-JEPA is to bridge the gap between simultaneous modeling of local trajectory details and global movement patterns by embedding explicit, cross-level trajectory abstractions into a JEPA framework. To that end, as Fig. 2 illustrates, given a trajectory $T$, we apply three consecutive convolutional layers followed by max pooling operations to produce point-level representation $T^{(1)}$, intermediate-level semantics $T^{(2)}$ and high-level summary $T^{(3)}$, where higher layer representations consist of coarser but semantically richer trajectory patterns. Trajectory abstraction at layer $l$ is learned by the corresponding JEPA layer $\text{JEPA}^{(l)}$ to capture multi-scale sequential dependencies.

**Spatial region representation.** Considering the continuous and high-precision nature of GPS coordinates, we partition the continuous spatial regions into fixed cells. But different from previous approaches Chang et al. (2023); Li et al. (2024b;a) that use grid cells, we employ Uber H3[1] to map GPS points into hexagonal grids to select the grid cell resolutions adaptively according to the study area size. Each hexagonal cell shares six equidistant neighbors, with all neighboring centers located at the same distance from the cell's center. Therefore, we structurally represent the spatial regions by

---

[1]https://h3geo.org/

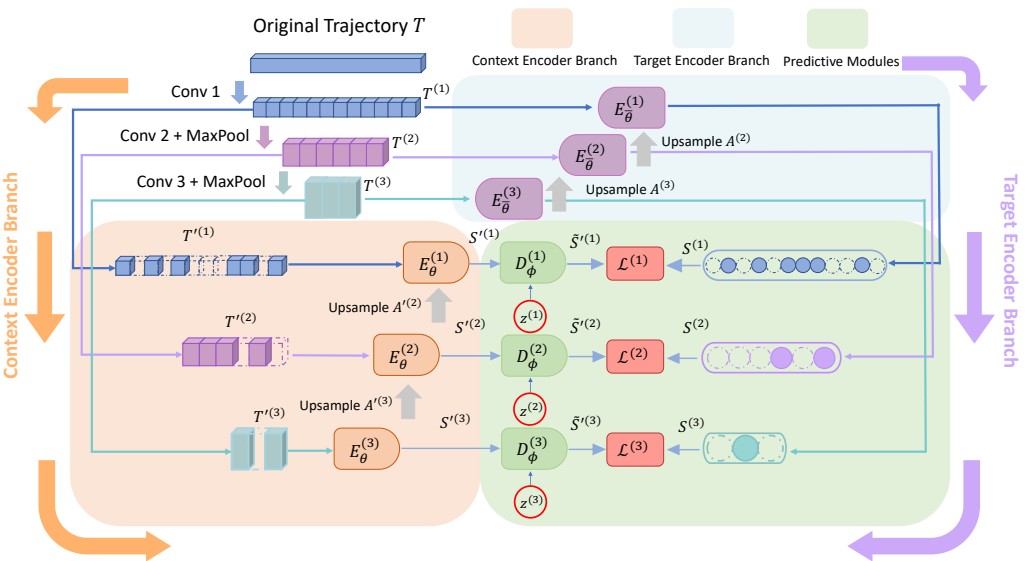

Figure 2: HiT-JEPA builds a three-level JEPA hierarchy to extract multi-scale trajectory semantics: (1) Level 1 encodes fine-grained, local point-level features; (2) Level 2 abstracts mesoscopic segment-level patterns; (3) Level 3 captures coarse, global route structures. Trajectory information is propagated from top to bottom, consecutive levels via attention weights.

a graph $\mathcal{G} = (V, E)$ where each node $v_i \in V$ is a hexagon cell connecting to its neighboring cells $v_j \in V$ by an undirected edge $e_{ij} \in E$. We pretrain the spatial node embeddings $\mathcal{H}$ of graph $\mathcal{G}$ using node2vec Grover & Leskovec (2016), which produces an embedding set:

$$\mathcal{H} = \big\{ h_i \in \mathbb{R}^d : v_i \in V \big\}, \tag{1}$$

where each $h_i$ encodes the relative position of node $v_i$. For a GPS location $P = (lon, lat)$, we first assign it to its grid cell index via:

$$\delta \colon \mathbb{R}^2 \to \{1, \ldots, |V|\}, \tag{2}$$

and then look up its embedding $h_{\delta(p)} \in \mathcal{H}$.

**Hierarchical trajectory abstractions.** After obtaining the location embeddings, we construct trajectory representations at multiple semantic levels, which are termed hierarchical trajectory abstractions. Given a trajectory $T$ with length $n$, we obtain its location embeddings and denote the input trajectory as $T = (h_{\delta(t_1)}, h_{\delta(t_2)}, \ldots, h_{\delta(t_n)}) \in (\mathbb{R}^d)^n$. Then, we create its multi-level abstractions $T^{(1)}$, $T^{(2)}$, $T^{(3)}$ by a set of convolutions with kernel size of 3 and stride of 2, and max pooling layers:

$$T^{(1)} = \text{LayerNorm}(\text{MaxPool1D}(\text{Conv1D}(T))) \in (\mathbb{R}^d)^{n_1}, \quad n_1 = n, \tag{3}$$

$$T^{(2)} = \text{LayerNorm}(\text{MaxPool1D}(\text{Conv1D}(T^{(1)}))) \in (\mathbb{R}^d)^{n_2}, \quad n_2 = \left\lfloor \frac{n_1}{2} \right\rfloor, \tag{4}$$

$$T^{(3)} = \text{LayerNorm}(\text{MaxPool1D}(\text{Conv1D}(T^{(2)}))) \in (\mathbb{R}^d)^{n_3}, \quad n_3 = \left\lfloor \frac{n_2}{2} \right\rfloor. \tag{5}$$

where $T^{(1)}$ in layer 1 preserves the channel dimension $d$ and sequence length $n_1 = n$, $T^{(2)}$ in layer 2 keeps the channel dimension and halves the sequence length to $n_2 = n/2$, and $T^{(3)}$ in layer 3 also keeps the channel dimension and halves the sequence length to $n_3 = n/4$. Higher-layer trajectory abstractions contain aggregated, high-level trajectory semantic behaviors, while lower layers preserve fine-grained, local dynamic details.

**Target encoder branch.** For the target encoder branch, at each level $l \in \{1, 2, 3\}$ the target trajectory representation is extracted by:

$$S^{(l)} = E_{\bar{\theta}}^{(l)}(T^{(l)}) \tag{6}$$

where $E_{\theta}^{(l)}$ is the target encoder at layer $l$. Similar to previous JEPA methods LeCun (2022); Assran et al. (2023); Li et al. (2024b); Bardes et al. (2023), we randomly sample $M$ times from target

representation to create the targets, where $S^{(l)}(i) = \{S_j^{(l)}\}_{j \in \mathcal{M}_i}$. Therefore, $S^{(l)}(i)$ is the $i$-th sampled target and $\mathcal{M}_i$ is the $i$-th sampling mask starting from a random position. To ensure the diversity of learning targets, we follow T-JEPA Li et al. (2024b) and introduce a set of masking ratios $r = \{r_1, r_2, r_3, r_4, r_5\}$ where each ratio value specifies the fraction of the representation to mask. At each sampling step, we uniformly draw one ratio from r. We also introduce a probability $p$: with probability $p$, we apply successive masking, and with probability $1 - p$, we scatter the masks randomly. Successive masking encourages the encoder to learn both local and long-range dependencies.

**Context encoder branch.** For the context encoder branch, we initially sample a trajectory context $C^{(l)}$ from $T^{(l)}$ at level $l$ by a mask $\mathcal{C}_T$ at with sampling ratio $p_\gamma$. Next, to prevent any information leakage, we remove from $C^{(l)}$ all positions that overlap with the targets $S^{(l)}$ to obtain the context input $T'^{(l)}$. The context trajectory representation $S'^{(l)}$ at level $l$ is extracted by:

$$S'^{(l)} = E_\theta^{(l)}(T'^{(l)}) \tag{7}$$

where $E_\theta^{(l)}$ is the context encoder at level $l$. During inference, we use $S'^{(1)}$ from $E_\theta^{(1)}$, enriched by the full hierarchy of multi-scale abstractions, as the final output of trajectory representations for similarity comparison or downstream fine-tuning.

**Predictions.** Once we have both context representations $S'^{(l)}$ and targets $S^{(l)}$ at level $l$, we apply JEPA predictor $D_\phi^{(l)}$ on $S'^{(l)}$ to approximate $S^{(l)}$ with the help of the mask tokens $z^{(l)}$:

$$\widetilde{S}'^{(l)}(i) = D_\phi^{(l)}(\text{CONCAT}(S'^{(l)}, \text{PE}(i) \oplus (z^{(l)}))) \tag{8}$$

where $\text{CONCATE}(\cdot)$ denotes concatenation and $\text{PE}(i)$ refers to the positional embedding after applying the target sampling mask $\mathcal{M}_i$. $\oplus$ is element-wise addition between these masked positional embeddings and the mask tokens. Then, we concatenate the mask tokens with positional information with the context representations to guide the predictor in approximating the missing components in the targets at the representation space.

**Hierarchical interactions.** By applying JEPA independently at each level, we learn trajectory representations at multiple scales of abstractions. However, the encoders at each level remain siloed and retain only their scale-specific information without leveraging insights from other layers. To enable hierarchical and multi-scale feature extraction, we propagate high-level information down to the next lower abstraction layer.

We adopt Transformer encoders Vaswani et al. (2017) for both context and target encoders as their self-attention module is proven highly effective in sequential modeling. Therefore, for both branches, we inject attention weights to the next lower level as a "top-down spotlight" where the high-level encoder casts its attention maps to the lower layer, lighting up where the lower-level encoder should attend. For clarity, we illustrate the process using the target encoder branch as an example. At level $l$, given the query and key matrices $Q^{(l)}$ and $K^{(l)}$ of an input trajectory abstraction $T^{(l)}$, we first retrieve the attention coefficient by:

$$d_k = \frac{d^{(l)}}{H}, Q_i^{(l)} = Q^{(l)} W_i^{Q,(l)}, \ K_i^{(l)} = K^{(l)} W_i^{K,(l)}, \ A_i^{(l)} = \text{softmax}\left(\frac{Q_i^{(l)} K_i^{(l)\top}}{\sqrt{d_k}}\right), \ i = 1, \ldots, H \tag{9}$$

where $H$ is the number of attention heads, $W_i^{Q,(l)}$ and $W_i^{K,(l)}$ are head-$i$ projections, $d^{(l)}$ is the channel dimension, and $A_i^{(l)}$ is the attention coefficient of the head-$i$. The multi-head attention coefficient $A^{(l)}$ are concatenated and projected by:

$$A^{(l)} = \text{Concat}\left(A_1^{(l)}, \ldots, A_H^{(l)}\right) W^{O,(l)} \tag{10}$$

where $W^{O,(l)}$ is the multi-head projection. To construct the output representation $S^{(l)}$ at level $l$, we simply apply the value matrix $V^{(l)}$ by:

$$S^{(l)} = A^{(l)} V^{(l)} \tag{11}$$

Since the dimension of $A^{(l)}$ is:

$$A^{(l)} \in [0,1]^{n^{(l)} \times n^{(l)}}, \quad n^{(l)} = \left\lfloor \frac{n^{(l-1)}}{2} \right\rfloor \tag{12}$$

where $n^{(l)}$ is the length of trajectory abstractions at level $l$, which is half of $n^{(l-1)}$ at level $l-1$ due to Eq. 4 and Eq. 5. We need to upsample the attention coefficients:

$$\widetilde{A}^{(l)} = \text{Interpolate}_{\text{bilinear}}\big(A^{(l)}\big) \in [0,1]^{n^{(l-1)} \times n^{(l-1)}} \tag{13}$$

where we adopt bilinear interpolation to upsample the attention weights at level $l$. To propagate the upsampled $\widetilde{A}^{(l)}$ to the next lower level, We refer to Chang et al. (2023) to calculate a weighted sum between $\widetilde{A}^{(l)}$ and lower level attention coefficient $A^{(l-1)}$. Therefore, we obtain the updated attention coefficient $A^{(l-1)}$ at level $l-1$ by:

$$A^{(l-1)} = (\sigma A^{(l-1)} + (1-\sigma)\widetilde{A}^{(l)}) \tag{14}$$

where $\sigma$ is a learnable scale factor weighting the importance of $A^{(l)}$. Attention coefficient $A'^{(l)}$ from the context encoders follows an identical procedure. This way, the coarse, global insights guide the fine-grained feature extraction in the next layer to focus on the most semantically important trajectory segments. This alignment sharpens local feature extraction so it stays consistent with the overall context.

**Loss function.** After obtaining the predicted representation $\widetilde{S}'^{(l)}(i)$ and the $i$-th target representation $S^{(l)}(i)$ at level $l$, we apply $\text{SmoothL1}$ to calculate the loss $\mathcal{L}^{(l)}$ between them:

$$\mathcal{L}^{(l)} = \underbrace{\frac{1}{MB} \sum_{i=1}^{M} \sum_{b=1}^{B} \sum_{n=1}^{N^{(l)}} \sum_{k=1}^{d^{(l)}} \text{SmoothL1}\big(\widetilde{S}'^{(l)}(i)_{b,n,k}, S^{(l)}(i)_{b,n,k}\big)}_{\mathcal{L}_{\text{JEPA}}^{(l)}} \tag{15}$$

$$+ \underbrace{\text{VarLoss}\big(z_{\text{tar}}^{(l)}\big) + \text{VarLoss}\big(z_{\text{ctx}}^{(l)}\big) + \text{CovLoss}\big(z_{\text{tar}}^{(l)}\big) + \text{CovLoss}\big(z_{\text{ctx}}^{(l)}\big)}_{\mathcal{L}_{\text{VICReg}}^{(l)}}.$$

where we sum over the channel and sequence length dimension $d^{(l)}$ and $N^{(l)}$, and average over the batch and number of target masks dimension $B$ and $M$ to obtain JEPA loss $\mathcal{L}_{\text{JEPA}}^{(l)}$. We also add VICReg Bardes et al. (2021) to prevent representation collapse, yielding more discriminative representations. We obtain the regularization term $\mathcal{L}_{\text{VICReg}}^{(l)}$ by summing up the variance loss $\text{VarLoss}(\cdot)$ and covariance loss $\text{CovLoss}(\cdot)$ of both expanded context representation $z_{\text{ctx}}^{(l)} = \text{MLP}(S'^{(l)})$ and expanded target representation $z_{\text{tar}}^{(l)} = \text{MLP}(S^{(l)})$ via a single-layer MLP. Afterwards, $\mathcal{L}_{\text{VICReg}}^{(l)}$ is added to the loss $\mathcal{L}^{(l)}$ at level $l$.

For level $l \in \{1, 2, 3\}$, we calculate a weighted sum to obtain the final loss $\mathcal{L}$:

$$\mathcal{L} = \lambda * \mathcal{L}^{(1)} + \mu * \mathcal{L}^{(2)} + \nu * \mathcal{L}^{(3)} \tag{16}$$

where $\lambda$, $\mu$ and $\nu$ are the scale factors for loss at each level.

## 4 EXPERIMENTS

We conduct experiments on three real-world urban GPS trajectory datasets: Porto [2], T-Drive Yuan et al. (2011; 2010) and GeoLife Zheng et al. (2008; 2009; 2010), two FourSquare datasets: FourSquare-TKY and FourSquare-NYC Yang et al. (2014), and one vessel trajectory dataset: Vessel Tracking Data Australia, which we call "AIS(AU)" [3]. The dataset details can be found in Appendices A.1. We compare HiT-JEPA with the three most recent self-supervised methods on trajectory similarity computation: TrajCL Chang et al. (2023), CLEAR Li et al. (2024a) and T-JEPA Li et al. (2024b). The details of these methods are listed in Appendices A.2

### 4.1 QUANTITATIVE EVALUATION

In this section, we evaluate HiT-JEPA and compare it to baselines in three experiments: most similar trajectory search, robustness of learn representations, and generalization with downstream fine-tuning. We combine the first two experiments as "Self-similarity".

---

[2]https://www.kaggle.com/c/pkdd-15-predict-taxi-service-trajectory-i/data
[3]https://www.operations.amsa.gov.au/spatial/DataServices/DigitalData

#### 4.1.1 SELF-SIMILARITY

Following similar experimental settings of previous work Chang et al. (2023); Li et al. (2024b), we construct a Query trajectory set $Q$ and a database trajectory $D$ for the testing set given a trajectory. $Q$ has 1,000 trajectories for Porto, T-Drive, and GeoLife, 600 for TKY, 140 for NYC, and 1400 for AIS(AU). And $D$ has 100,000 trajectories for Porto, 10,000 for T-Drive and Geolife, 3000 for TKY, 700 for NYC, and 7000 for AIS(AU). Detailed experimental settings can be found in Appendices A.4.

Table 1: Mean-rank comparison of methods across meta ratios $R_1 \sim R_5$. For each meta ratio, we report the mean ranks under varying DB size $|D|$, downsampling rate $\rho_s$, and distortion rate $\rho_d$. **Bold** value are the lowest mean ranks and underlined values are the second lowest.

| Dataset | Method | $R_1$ | | | $R_2$ | | | $R_3$ | | | $R_4$ | | | $R_5$ | | |
|---|---|---|---|---|---|---|---|---|---|---|---|---|---|---|---|---|
| | | $|D|$ | $\rho_s$ | $\rho_d$ | $|D|$ | $\rho_s$ | $\rho_d$ | $|D|$ | $\rho_s$ | $\rho_d$ | $|D|$ | $\rho_s$ | $\rho_d$ | $|D|$ | $\rho_s$ | $\rho_d$ |
| Porto | TrajCL | **1.004** | **1.047** | **1.017** | **1.007** | **1.170** | **1.029** | **1.008** | **1.905** | **1.036** | **1.011** | 6.529 | **1.060** | **1.014** | 68.557 | **1.022** |
| | TrjSR | 3.240 | 12.553 | 12.509 | 5.321 | 16.945 | 15.401 | 7.073 | 37.150 | 15.901 | 8.740 | 65.413 | 28.914 | 10.192 | 149.950 | 32.730 |
| | CLEAR | 3.235 | 7.796 | 4.250 | 4.012 | 13.323 | 4.442 | 4.088 | 22.814 | 4.284 | 4.137 | 44.865 | 4.438 | 4.204 | 123.921 | 4.399 |
| | T-JEPA | 1.029 | 1.455 | 1.097 | 1.048 | 2.304 | 1.084 | 1.053 | 4.413 | 1.115 | 1.061 | 9.599 | 1.110 | 1.074 | **23.900** | 1.123 |
| | HiT-JEPA | 1.026 | 1.369 | 1.074 | 1.043 | 2.624 | 1.077 | 1.048 | 5.541 | 1.085 | 1.058 | 13.773 | 1.093 | 1.065 | 28.806 | 1.119 |
| T-Drive | TrajCL | 1.111 | 1.203 | 1.267 | 1.128 | 1.348 | 3.320 | 1.146 | 1.668 | 1.355 | 1.177 | **1.936** | 1.513 | 1.201 | 3.356 | 1.179 |
| | TrjSR | 110.726 | 674.16 | 581.776 | 223.841 | 795.331 | 572.944 | 356.941 | 870.73 | 566.816 | 475.872 | 960.404 | 545.278 | 592.146 | 1033.404 | 566.696 |
| | CLEAR | 1.047 | 1.305 | 1.111 | 1.062 | 1.484 | 1.110 | 1.077 | 1.964 | 1.171 | 1.088 | 3.497 | 1.152 | 1.104 | 3.902 | 1.172 |
| | T-JEPA | **1.032** | 1.088 | 1.054 | **1.034** | 1.225 | 1.061 | **1.036** | 1.617 | 1.069 | 1.045 | 3.226 | 1.067 | 1.049 | 4.115 | 1.078 |
| | HiT-JEPA | 1.040 | **1.057** | **1.035** | 1.040 | **1.085** | **1.029** | 1.040 | **1.172** | **1.039** | **1.040** | 1.389 | **1.033** | **1.040** | **2.222** | **1.034** |
| GeoLife | TrajCL | 1.130 | 1.440 | 7.973 | 1.168 | 1.435 | 19.266 | 1.195 | 1.720 | 12.397 | 1.234 | 1.616 | 10.560 | 1.256 | 2.675 | 11.035 |
| | TrjSR | 6.765 | 8.332 | 7.747 | 7.393 | 8.594 | 7.942 | 7.661 | 8.688 | 7.648 | 7.767 | 8.566 | 8.534 | 8.350 | 8.770 | 9.460 |
| | CLEAR | 1.110 | 1.196 | 1.212 | 1.124 | 1.318 | 1.211 | 1.144 | 1.818 | 1.189 | 1.155 | 2.237 | 1.239 | 1.155 | 3.712 | 1.333 |
| | T-JEPA | **1.019** | **1.052** | **1.047** | 1.034 | **1.030** | **1.093** | 1.036 | **1.103** | **1.101** | 1.040 | **1.150** | **1.154** | 1.047 | **1.218** | **1.197** |
| | HiT-JEPA | 1.033 | 1.061 | 1.170 | **1.033** | 1.111 | 1.370 | **1.033** | 1.247 | 1.357 | **1.033** | 1.377 | 1.509 | **1.033** | 1.573 | 1.511 |
| TKY (zero-shot) | TrajCL | 17.590 | 66.963 | 75.397 | 32.377 | 67.835 | 79.228 | 46.958 | 116.677 | 59.222 | 62.145 | 170.460 | 69.642 | 78.722 | 211.487 | 65.258 |
| | TrjSR | 8.673 | 31.770 | 27.505 | 17.120 | 37.070 | 30.758 | 22.310 | 48.985 | 30.923 | 26.820 | 64.380 | 33.113 | 29.318 | 84.302 | 34.043 |
| | CLEAR | 119.561 | 591.345 | 583.863 | 242.493 | 626.075 | 591.460 | 349.132 | 646.160 | 587.138 | 456.525 | 662.553 | 588.212 | 577.238 | 709.903 | 591.107 |
| | T-JEPA | 1.948 | 3.060 | 3.245 | 2.272 | 4.227 | 3.165 | 2.617 | 7.975 | 3.313 | 2.913 | 18.173 | 3.202 | 3.275 | 19.135 | 3.127 |
| | HiT-JEPA | **1.508** | **2.490** | **2.060** | **1.707** | **2.962** | **2.002** | **1.835** | **4.985** | **2.067** | **1.930** | **10.268** | **2.045** | **2.057** | **14.755** | **1.988** |
| NYC (zero-shot) | TrajCL | 4.336 | 16.886 | 15.093 | 6.457 | 18.857 | 16.971 | 9.129 | 22.007 | 16.443 | 12.350 | 37.579 | 11.236 | 15.071 | 36.650 | 6.543 |
| | TrjSR | 3.929 | 5.457 | 6.307 | 4.793 | 5.171 | 7.950 | 5.457 | 8.350 | 6.679 | 5.821 | 12.757 | 7.443 | 6.007 | 14.329 | 7.907 |
| | CLEAR | 19.693 | 68.843 | 68.057 | 32.171 | 74.964 | 68.321 | 43.214 | 75.121 | 69.221 | 55.507 | 79.514 | 70.507 | 67.207 | 84.421 | 65.914 |
| | T-JEPA | 1.450 | 1.950 | 1.714 | 1.514 | 3.050 | 1.736 | 1.571 | 2.400 | 1.679 | 1.636 | 2.457 | 1.771 | 1.714 | 5.850 | 1.807 |
| | HiT-JEPA | **1.343** | **1.743** | **1.493** | **1.364** | **2.143** | **1.500** | **1.414** | **1.636** | **1.500** | **1.457** | **2.407** | **1.550** | **1.500** | **3.343** | **1.471** |
| AIS(AU) (zero-shot) | TrajCL | 9.057 | 37.721 | 37.866 | 18.771 | 9.878 | 37.879 | 26.538 | 41.068 | 37.862 | 33.004 | 45.352 | 37.911 | 37.866 | 48.651 | 38.399 |
| | TrjSR | 692.000 | 3658.400 | 3649.450 | 1390.364 | 3661.421 | 3649.407 | 2136.271 | 3675.043 | 3649.150 | 2942.586 | 3714.564 | 3649.086 | 2892.264 | 3700.221 | 3649.371 |
| | CLEAR | 38.042 | 188.171 | 184.600 | 73.164 | 187.914 | 184.579 | 112.371 | 192.571 | 184.600 | 150.050 | 191.629 | 184.871 | 184.600 | 198.843 | 184.593 |
| | T-JEPA | 2.156 | 5.661 | 4.753 | 3.176 | 6.849 | 4.753 | 3.889 | **9.486** | 4.755 | 4.364 | 13.055 | 4.758 | 4.754 | 16.986 | 4.749 |
| | HiT-JEPA | **1.483** | **4.119** | **2.759** | **1.954** | **6.357** | **2.759** | **2.311** | 10.233 | **2.758** | **2.579** | **15.180** | **2.757** | **2.758** | **20.267** | **2.755** |

Table 1 shows the mean ranks of all methods. HiT-JEPA achieves the overall lowest mean ranks across five of the six datasets. For urban GPS datasets, Porto, T-Drive, and GeoLife, we have the lowest ranks in the T-Drive dataset. For example, the mean ranks of DB size $|D|$ across 20%~100% and distortion rates $\rho_d$ across 0.1~0.5 remains very steady (1.040~1.041 and 1.031~1.038). This dataset has taxi trajectories with much longer irregular sampling intervals (3.1 minutes on average). By leveraging a hierarchical structure to capture the global and high-level trajectory abstractions, HiT-JEPA learns features that remain invariant against noise and sparse sampling, resulting in more robust and accurate representations against low and irregularly sampled trajectories with limited training samples. We achieve comparative mean ranks (only 2.8% higher) with T-JEPA on GeoLife, and overall, the second best on Porto. This is because Porto trajectories inhabit an especially dense spatial region, so TrajCL can exploit auxiliary cues such as movement speed and orientations to tease apart nearly identical paths. However, relying on these features undermines the generalization ability in lower-quality trajectories (e.g., in T-Drive) and knowledge transfer into other cities.

Next, we evaluate zero-shot performance on TKY, NYC, and AIS(AU). HiT-JEPA consistently achieves the lowest mean ranks across all database sizes, downsampling, and distortion rates. Both TKY and NYC consist of highly sparse and coarse check-in sequences, lacking trajectory waypoints, which challenge the summarization ability of the models. Benefiting from the hierarchical structure, HiT-JEPA first summarizes the mobility patterns at a coarse level, then refines the check-in details at finer levels. Crucially, the summarization knowledge is transferred from dense urban trajectories in Porto, demonstrating that HiT-JEPA learns more generalizable representations than TrajCL in Porto with more essential spatiotemporal information captured in trajectories. Even on AIS(AU) with trajectories across the ocean-wide scales, HiT-JEPA maintains overall the lowest mean ranks, demonstrating its ability to handle multiple forms of trajectories that spread over various regional scales. We find that even though CLEAR outperforms TrajCL on T-Drive and GeoLife, it exhibits weak generalization in zero-shot experiments on TKY, NYC, and AIS(AU). TrjSR showed the weakest overall perfor-

Table 2: Comparisons with fine-tuning 2-layer MLP decoder. **Bold** value are the lowest mean ranks and underlined values are the second lowest.

| Dataset | Method | EDR | | | LCSS | | | Hausdorff | | | Fréchet | | | Average |
|---|---|---|---|---|---|---|---|---|---|---|---|---|---|---|
| | | HR@5↑ | HR@20↑ | R5@20↑ | HR@5↑ | HR@20↑ | R5@20↑ | HR@5↑ | HR@20↑ | R5@20↑ | HR@5↑ | HR@20↑ | R5@20↑ | |
| Porto | TrajCL | 0.137 | 0.179 | 0.301 | 0.329 | 0.508 | 0.663 | 0.456 | 0.574 | 0.803 | 0.412 | 0.526 | 0.734 | 0.468 |
| | TrjSR | 0.085 | 0.083 | 0.157 | 0.162 | 0.197 | 0.292 | 0.166 | 0.192 | 0.304 | 0.157 | 0.173 | 0.288 | 0.188 |
| | CLEAR | 0.078 | 0.075 | 0.142 | 0.164 | 0.198 | 0.293 | 0.152 | 0.131 | 0.232 | 0.192 | 0.165 | 0.316 | 0.178 |
| | T-JEPA | 0.154 | 0.194 | 0.336 | 0.365 | 0.551 | 0.713 | **0.525** | **0.633** | **0.869** | 0.433 | 0.565 | 0.771 | **0.509** |
| | HiT-JEPA | **0.163** | **0.197** | **0.337** | **0.369** | **0.558** | **0.720** | 0.466 | 0.599 | 0.835 | **0.450** | **0.587** | **0.810** | 0.508 |
| T-Drive | TrajCL | 0.094 | 0.131 | 0.191 | 0.159 | 0.289 | 0.366 | 0.173 | 0.256 | 0.356 | 0.138 | 0.187 | 0.274 | 0.218 |
| | TrjSR | 0.076 | 0.068 | 0.114 | 0.076 | 0.080 | 0.118 | 0.095 | 0.090 | 0.143 | 0.098 | 0.094 | 0.145 | 0.100 |
| | CLEAR | 0.093 | 0.084 | 0.143 | 0.126 | 0.166 | 0.216 | 0.142 | 0.158 | 0.243 | 0.135 | 0.170 | 0.283 | 0.163 |
| | T-JEPA | 0.094 | 0.147 | 0.215 | 0.205 | 0.366 | 0.469 | 0.158 | 0.229 | 0.329 | 0.125 | 0.159 | 0.249 | 0.229 |
| | HiT-JEPA | **0.112** | **0.170** | **0.260** | **0.221** | **0.384** | **0.493** | **0.222** | **0.316** | **0.456** | **0.158** | **0.219** | **0.325** | **0.278** |
| GeoLife | TrajCL | 0.193 | 0.363 | 0.512 | 0.232 | 0.484 | 0.584 | 0.479 | 0.536 | 0.745 | 0.398 | 0.463 | 0.708 | 0.475 |
| | TrjSR | 0.138 | 0.246 | 0.443 | 0.229 | 0.330 | 0.479 | 0.492 | 0.439 | 0.692 | 0.383 | 0.362 | 0.614 | 0.404 |
| | CLEAR | 0.175 | 0.164 | 0.311 | 0.224 | 0.224 | 0.342 | 0.347 | 0.308 | 0.499 | 0.347 | 0.273 | 0.539 | 0.320 |
| | T-JEPA | **0.195** | 0.383 | 0.527 | 0.242 | 0.515 | 0.586 | 0.606 | 0.656 | 0.857 | **0.488** | 0.406 | 0.731 | 0.516 |
| | HiT-JEPA | 0.183 | **0.414** | **0.564** | **0.250** | **0.525** | **0.609** | **0.643** | **0.700** | **0.885** | 0.467 | **0.555** | **0.842** | **0.553** |

mance across all datasets. This is because image-based representations have difficulty distinguishing fine-grained trajectory differences, a challenge exacerbated by lower data quality (e.g., T-Drive).

### 4.1.2 DOWNSTREAM FINE-TUNING

To evaluate the generalization ability of HiT-JPEA, we conduct downstream fine-tuning on its learned representations. Specifically, we retrieve and freeze the encoder of HiT-JEPA and other baselines, concatenated with a 2-layer MLP decoder, then train the decoder to approximate the computed trajectory similarities by heuristic approaches. This setting is first proposed by TrajCL Chang et al. (2023), then followed by T-JEPA Li et al. (2024b), to quantitatively assess whether the learned representations can generalize to approach the computational processes underlying each heuristic measure. In real applications, fine-tuned models can act as efficient, "fast" approximations of traditional heuristic measures, alleviating their quadratic time-complexity bottleneck. We report hit ratios HR@5 and HR@20 to evaluate the correct matches between top-5 predictions and each of the top-5 and top-20 ground truths. We also report the recall R5@20 to evaluate the correct matches of top-5 ground truths from predicted top-20 predictions. We approximate all model representations to 4 heuristic measures: EDR, LCSS, Hausdorff and Discret Fréchet. We do not include TrjSR here as its results are proven to be less competitive in Chang et al. (2023).

From Table 2, we can observe that HiT-JEPA achieves the highest overall performance. In the column "Average", we calculate the average of all reported results for each model on each dataset. HiT-JEPA outperforms T-JEPA on T-Drive and GeoLife for 12.6% and 6.4%, with only 3.7% lower on Porto. For results on T-Drive, HiT-JEPA consistently outperforms the T-JEPA across all measures, especially in Hausdorff and Discret Fréchet measures, where we achieve relative average improvements of 14.7% and 19.9%, respectively. For GeoLife, even though we have some cases that achieve slightly lower results than T-JEPA in EDR and Hausdorff, we are overall 6.1% and 1.8% higher on average in these two measures. For Porto, although our results are 3.7% lower than T-JEPA on average across all measures, we have successfully made minor improvements in LCSS measure. Visualizations of predictions can be found in Fig. 12 and Fig. 13 in Appendices A.8.

### 4.2 VISUALIZATIONS OF HiT-JEPA EMBEDDINGS.

HiT-JEPA encodes and predicts trajectory information only in the representation space, making it more difficult than generative models such as MAE He et al. (2022) to evaluate the learned representation quality at the data level. To assess and gauge the validity of the representations of HiT-JEPA, we project the encoded $S'^{(1)}$ from $E_\theta^{(1)}$ (on full trajectories) and predicted $\widetilde{S}'^{(1)}$ from $D_\phi^{(1)}$ (on masked trajectories) back onto the hexagonal grid at their GPS coordinates for visual comparisons.

First, we freeze the context encoders and predictors across all levels in a pre-trained HiT-JEPA. Then we encode and predict the masked trajectory representations to simulate the training process, and encode the full trajectory representations to simulate the inference process. Next, we concatenate and tune a 2-layer MLP for each of the representations to decode to the hexagonal grid cell embeddings

to which they belong. We denote the decoded predicted masked trajectory representations as $S_1$ and the decoded encoded full trajectory representations as $S_2$. Finally, for each trajectory position, we search for the $k$ most similar embeddings in the spatial region embedding set $\mathcal{H}$ and retrieve their hexagonal cell IDs. We choose $k = 3$ in our visualizations.

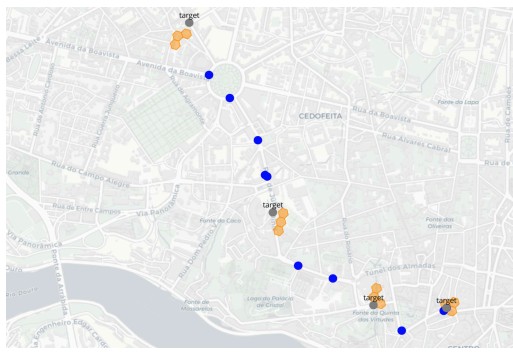 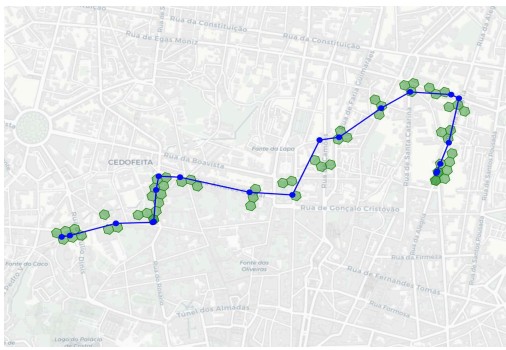

(a) Predicted masked points          (b) Encoded full trajectory

Figure 3: Visualizations of decoded learned trajectory representations by HiT-JEPA on hexagonal cells: (a) blue points are sampled trajectory points, gray points are masked trajectory points labeled with "target", and orange hexagons are projected predictions. (b) blue points are full trajectory points, green hexagons are projected encoded representations.

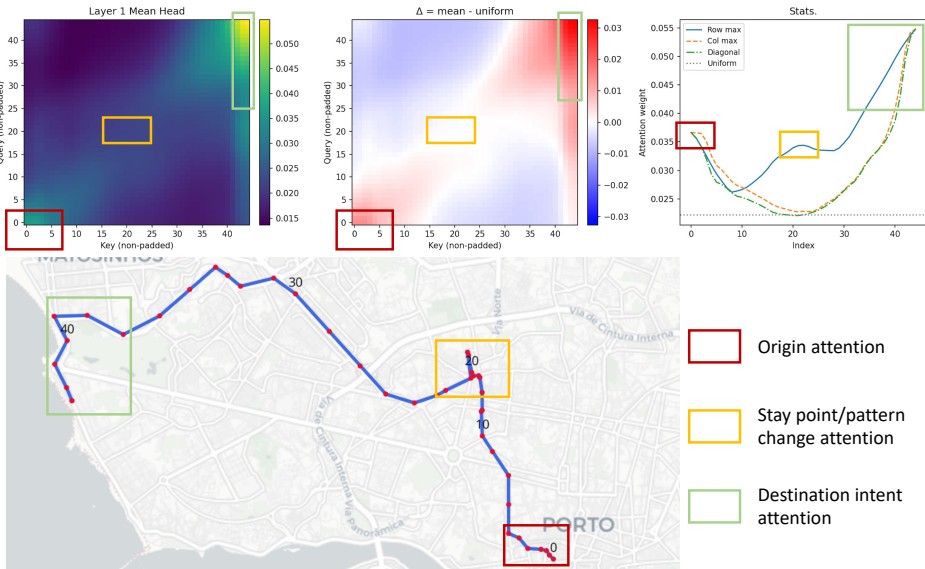

Figure 4: A case study of hierarchical semantic information captured by HiT-JEPA. **(Top-Left)** The raw attention map visualizes the absolute attention weights, showing the overall intensity distribution. **(Top-Mid)** The deviation heatmap by displaying areas of active focus (red) versus suppression (blue) relative to the mean attention value. **(Top-Right)** The statistical profiles quantify the peak attention intensity at each time step. **Bottom** The corresponding physical trajectory with index labeled every 10 steps, where colored boxes spatially ground the salient attention regions identified in the top row.

Fig. 3a shows the comparisons between decoded cells (orange hexagons) and masked points (gray points) labeled as "targets". The decoded locations lie in close proximity to their corresponding masked targets, confirming that the model effectively learns accurate representations for masked points during training. Fig. 3b overlays the decoded cells green hexagons) on each blue trajectory point, demonstrating that the model can encode each point with even greater accuracy with access to the full trajectory during inference.

## 4.3 INTERPRETATION OF HIERARCHICAL ATTENTION WEIGHTS.

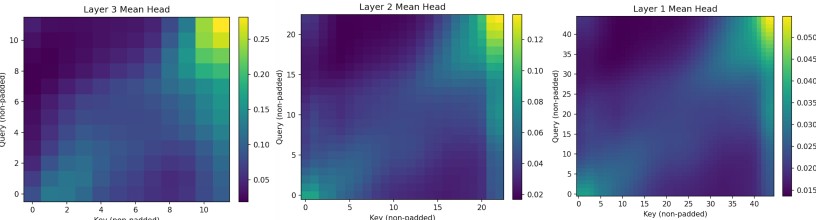

Figure 5: Averaged attention weight visualizations at each JEPA layer. Left to right: $A^{(3)}$ to $A^{(1)}$.

From Fig. 4, by corroborating the raw attention map, deviation heatmap, and statistical profiles, we identify three distinct semantic phases localized within the bottom-left, top-right, and middle regions around the $20^{th}$ trajectory point. These phases correspond to the peak intensity of attention allocated to specific trajectory segments: the origin anchoring (red boxes), the local attention peak triggered by a pattern change (orange boxes), and the destination intent (green boxes). This spatial-semantic alignment confirms that HiT-JEPA successfully learns the critical semantic waypoints from raw GPS tracks and verifies the interpretability of its learned representations. By comparing the raw attention weights across 3 JEPA layers in Fig. 5, it is obvious to discern a coarse-to-fine attention evolution, where the $A^3$ layer highly summarizes the trajectory origin-destination patterns and is fused into lower layers with more smoothed local details. This validates that HiT-JEPA learns consistent trajectory semantics through the hierarchical interactions while preserving distinct layer-specific granularity. We further visualize the attention map for each head in Appendices A.6.

### 4.4 ABLATION STUDY

We study the effect of removing the key designs in HiT-JEPA. We compare HiT-JEPA with 4 variants: 1) **HiT_emb** which replaces the hierarchical interaction method from attention upsampling to directly concatenate the upsampled encoder embeddings between $S'^{(l)}$ and $S'^{(l-1)}$. 2) **HiT_single_layer** where we only level $l = 1$ to train and predict. 3) **HiT_no_attn** with no hierarchical interactions between each pair of successive layers. 4) **HiT_rect** with spatial location tokenization method changed to rectangular grid cells. We train these variants and conduct self-similarity experiments on Porto.

Table 3 shows the comparisons between HiT-JEPA and its variants. The performance drops without any key designs, especially for HiT_emb, as directly concatenating the embedding from the previous layers causes representation collapse. Results from the other two variants demonstrate that in our model design, even though each layer of $\text{JEPA}^l$ can learn individually, the hierarchical interactions bind different levels into a cohesive multi-scale structure.

Table 3: Ablation Study of HiT-JEPA on Porto

| **Varying DB Size** $|D|$ | | | | |
|---|---|---|---|---|
| **Model** | **20%** | **40%** | **60%** | **80%** | **100%** |
| HiT_emb | 106.568 | 209.746 | 297.919 | 394.111 | 497.064 |
| HiT_single_layer | 1.031 | 1.061 | 1.066 | 1.077 | 1.091 |
| HiT_no_attn | 1.026 | 1.049 | 1.054 | 1.062 | 1.069 |
| HiT_rect | 1.032 | 1.062 | 1.069 | 1.080 | 1.093 |
| HiT-JEPA | **1.026** | **1.043** | **1.048** | **1.058** | **1.065** |
| **Downsampling Rate** $\rho_s$ | | | | |
| **Model** | **0.1** | **0.2** | **0.3** | **0.4** | **0.5** |
| HiT_emb | 569.322 | 706.831 | 1004.246 | 2047.699 | 2171.331 |
| HiT_single_layer | 1.378 | 2.659 | 5.626 | 14.123 | **26.875** |
| HiT_no_attn | 1.405 | 2.867 | 5.761 | 17.143 | 27.324 |
| HiT_rect | 1.508 | 3.054 | 7.735 | 18.912 | 36.768 |
| HiT-JEPA | **1.369** | **2.624** | **5.541** | **13.773** | 28.806 |
| **Distortion Rate** $\rho_d$ | | | | |
| **Model** | **0.1** | **0.2** | **0.3** | **0.4** | **0.5** |
| HiT_emb | 502.259 | 503.876 | 506.333 | 507.738 | 507.082 |
| HiT_single_layer | 1.088 | 1.099 | 1.120 | 1.100 | 1.137 |
| HiT_no_attn | 1.079 | 1.095 | 1.105 | 1.093 | 1.120 |
| HiT_rect | 1.095 | 1.111 | 1.123 | 1.122 | 1.124 |
| HiT-JEPA | **1.074** | **1.077** | **1.085** | **1.093** | **1.119** |

### 5 CONCLUSION

In summary, HiT-JEPA introduces a unified three-layer hierarchy that captures point-level fine-grained details, intermediate trajectory patterns, and high-level trajectory semantics within a single self-supervised framework. By leveraging a Hierarchical JEPA, it enables a more powerful trajectory feature extraction in the representation space and produces cohesive multi-granular embeddings. Extensive evaluations on diverse urban and maritime trajectory datasets show that HiT-JEPA outperforms single-scale self-supervised methods in trajectory similarity computation, especially zero-shot generalization and downstream fine-tuning. These results validate its effectiveness and robustness for real-world, large-scale trajectory modeling.

## 6 ETHICS STATEMENT

We claim that we adhere to the ICLR Code of Ethics. All the datasets used in the manuscript are publicly available with no user information revealed. HiT-JEPA encodes the trajectory location information in hexagonal cell tokens, where exact GPS traces are blurred. And such tokens are the only input to our model, thereby preventing any leakage of precise location data. The code for all baselines is publicly available and used under their respective licenses.

## 7 REPRODUCIBILITY STATEMENT

We provide an anonymous GitHub link `https://anonymous.4open.science/r/HiT-JEPA` to prove that our work is reproducible. This repository contains the code for the HiT-JEPA method implementation in Section 3 and any data processing and evaluation files in Section 4. The details, such as dataset statistics A.1, implementation configurations A.3, and further experiment details A.4, can also be found in the repository.

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

# A APPENDIX

## A.1 DATASETS

Here we list the details of the datasets:

- **Porto** includes 1.7 million trajectories from 442 taxis in Porto, Portugal. The dataset was collected from July 2013 to June 2014.
- **T-Drive** contains trajectories of 10,357 taxis in Beijing, China from Feb. 2 to Feb. 8, 2008. The average sampling interval is 3.1 minutes.
- **GeoLife** contains trajectories of 182 users in Beijing, China from April 2007 to August 2012. There are 17,6212 trajectories in total with most of them sampled in 1–5 seconds.
- **Foursquare-TKY** is collected for 11 months from April 2012 to February 2013 in Tokyo, Japan, with 573,703 check-ins in total.
- **Foursquare-NYC** is collected for 11 months from April 2012 to February 2013 in New York City, USA, with 227,428 check-ins in total.
- **AIS(AU)** comprises vessel traffic records collected by the Craft Tracking System (CTS) of Australia. In this paper, we use vessel trajectories in February 2025.

Table 4: Statistics of Datasets after preprocessing.

| Data type | Dataset | #points | #trajectories |
|---|---|---|---|
| Urban trajectories | Porto | 65,913,828 | 1,372,725 |
| | T-Drive | 5,579,067 | 101,842 |
| | GeoLife | 8,987,488 | 50,693 |
| Check-in sequences | TKY | 106,480 | 3,048 |
| | NYC | 28,858 | 734 |
| Vessel trajectories | AIS(AU) | 485,424 | 7,095 |

We first keep trajectories in urban areas with the number of points ranging from 20 to 200, where the statistics of the datasets after preprocessing are shown in Table 4. We use 200,000 trajectories for Porto, 70,000 for T-Drive, and 35000 for GeoLife as training sets. Each dataset has 10% of data used for validation. As there are many fewer trajectories in TKY, NYC, and AIS(AU), we use all trajectories in these datasets for testing. For the testing set, we select 100,000 trajectories for Porto, 10,000 for T-Drive and GeoLife, 3000 for TKY, 700 for NYC, and 7000 for AIS(AU). For the downstream fine-tuning task, we select 10,000 trajectories for Porto and T-Drive, and 5000 for GeoLife, where the selected trajectories are split by 7:1:2 for training, validation, and testing. We train Hit-JEPA and all baselines from scratch for Porto, T-Drive, and GeoLife datasets. Then, we load the pre-trained weights from Porto and conduct zero-shot self-similarity experiments on each of the TKY, NYC, and AIS(AU) to evaluate the generalization ability of all models.

## A.2 BASELINES

We compare HiT-JEPA with four most recent self-supervised free space trajectory similarity computation methods: TrajCL Chang et al. (2023), TrjSR Cao et al. (2021), CLEAR Li et al. (2024a), and T-JEPA Li et al. (2024b). TrajCL is a contrastive learning method that adopts a dual-feature attention module to capture the trajectory details, which has achieved impactful performance on trajectory similarity computation in multiple datasets and experimental settings. TrjSR is a generative model that converts trajectories into gray-scale images. This method reconstructs the high-resolution trajectory image from the low-resolution image by leveraging single-image super-resolution to learn better spatial trajectory representations. CLEAR improves the contrastive learning process by ranking the positive trajectory samples based on their similarities to anchor samples, capturing detailed differences from similar trajectories. T-JEPA is the most recent method utilizing Joint Embedding Predictive Architecture to encode and predict trajectory information in the representation space, which effectively captures necessary trajectory information. We run these two models from their open-source code repositories with default parameters.

### A.3 IMPLEMENTATION DETAILS

We use Adam Optimizer for training and optimizing the model parameters across all levels, except for the target encoders. The target encoder at each level $l$ updates its parameters via the exponential moving average of the parameters of the context encoder at the same level. The maximum number of training epochs is 20, and the learning rate is 0.0001, decaying by half every 5 epochs. The embedding dimension $d$ is 256, and the batch size is 64. We apply 1-layer Transformer Encoders for both context and target encoders at each level, with the number of attention heads set to 8 and hidden layer dimension to 1024. We use a 1-layer Transformer Decoder as the predictor at each level $l$ with the number of attention heads set to 8. We use learnable positional encoding for all the encoders and decoders. We set the resampling masking ratio to be selected from $r = \{10\%, 15\%, 20\%, 25\%, 30\%\}$ and the number of sampled targets $M$ to 4 for each trajectory at each model level $l$. The successive sampling probability $p$ is set to 50%, and the initial context sampling ratio $p_\gamma$ is set to range from 85% to 100%. The scale factors for the final loss are $\lambda = 0.05$, $\mu = 0.15$, and $\nu = 0.8$. We use a hexagonal cell resolution of 11 for Porto, resolution 10 for T-Drive, GeoLife, TKY, and NYC, and resolution 4 for AIS(AU). All experiments are conducted on servers with Nvidia A5000 GPUs, 24GB of memory, and 250GB of RAM.

### A.4 EXPERIMENTAL SETTINGS

#### A.4.1 SELF-SIMILARITY

For each query trajectory $q \in Q$, we create two sub-trajectories $q_a = \{p_1, p_3, p_5, \ldots\}$ containing the odd-indexed points and $q_b = \{p_2, p_4, p_6, \ldots\}$ even-indexed points of $q$. We separate them by putting $q_a$ into the query set $Q$ and putting $q_b$ into the database $D$, with the rest of the trajectories in $D$ randomly filled from the testing set. Each $q_a$ and $q_b$ pair exhibits similar overall patterns in terms of shape, length, and sampling rate. We apply HiT-JEPA context-encoders to both query and database trajectories, compute pairwise similarities, and sort the results in descending order. Next, we report the mean rank of each $q_b$ when retrieved by its corresponding query $q_a$: ideally, the true match appears at rank one. We choose $\{20\%, 40\%.60\%, 80\%, 100\%\}$ of the total database size $|D|$ for evaluation. To further evaluate the robustness of learned trajectory representations, we also apply down-sampling and distortion on $Q$ and $D$. Specifically, we randomly mask points (with start and end points kept) with down-sampling probability $\rho_s$ and shift the point coordinates with distortion probability $\rho_d$. Both $\rho_s$ and $\rho_d$ represent the number of points to be down-sampled or distorted, ranging from $\{0.1, 0.2, 0.3, 0.4, 0.5\}$.

For the convenience of comparing results under these settings together, we denote meta ratio $R_i = \{|D|_i, \rho_{si}, \rho_{di}\}$ and compare the **mean rank** of all models at each $R_i$ on each dataset, smaller values are better.

### A.5 HYPERPARAMETER ANALYSIS

We analyze the impact of two sets of hyperparameters with the implementation and experimental settings in the Appendices section A.3 and A.4.

**Number of attention layers at each abstraction level.** We change the number of Transformer encoder layers for each level to 2 and 3, then compare them with the default setting (1 layer) for self-similarity search with varying $|D|$, $\rho_s$ and $\rho_d$ on Porto. From Fig. 6, we can find that with only 1 attention layer, we can achieve the lowest mean ranks for all settings. This is due to higher chances of overfitting with more attention layers.

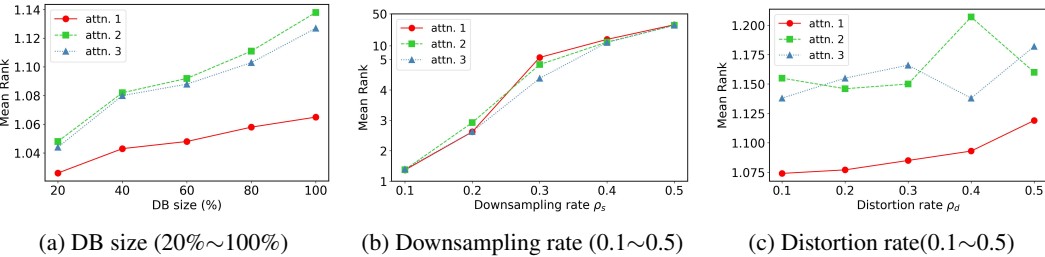

(a) DB size (20%∼100%)          (b) Downsampling rate (0.1∼0.5)          (c) Distortion rate(0.1∼0.5)

Figure 6: Effect of different numbers of attention layers at each abstraction level.

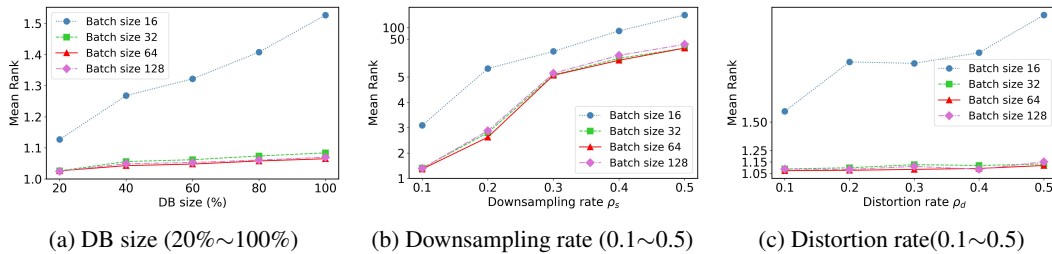

(a) DB size (20%∼100%)          (b) Downsampling rate (0.1∼0.5)          (c) Distortion rate(0.1∼0.5)

Figure 7: Effect of different batch sizes.

**Weighting coefficient for the multi-level loss** $\mathcal{L}$**.** The values of the loss weighting coefficients $\lambda$, $\mu$, and $\nu$ are carefully tuned. In Table 5, 6, and 7, we compare our selected coefficients with other 3 sets of parameters in a wide range on the Porto dataset. From the tables, we can see that HiT-JEPA is robust against various loss combinations. Even though the loss coefficients with $\lambda$, $\mu$, and $\nu$ equal to 0.33, 0.33, and 0.33 perform better on the downsampling experiment, our selected combination with $\lambda = 0.05$, $\mu = 0.15$, and $\nu = 0.8$ still learns overall the most accurate, stable, and consistent results across all experimental settings.

| $\lambda$ | $\mu$ | $\nu$ | 20% | 40% | 60% | 80% | 100% |
|---|---|---|---|---|---|---|---|
| 0.1 | 0.2 | 0.7 | 1.026 | 1.050 | 1.056 | 1.067 | 1.079 |
| 0.33 | 0.33 | 0.33 | 1.036 | 1.072 | 1.080 | 1.102 | 1.120 |
| 0.6 | 0.3 | 0.1 | 1.035 | 1.063 | 1.066 | 1.079 | 1.099 |
| **0.05** | **0.15** | **0.8** | **1.026** | **1.043** | **1.048** | **1.058** | **1.065** |

Table 5: Loss weighting coefficients for varying DB sizes $|D|$.

| $\lambda$ | $\mu$ | $\nu$ | 0.1 | 0.2 | 0.3 | 0.4 | 0.5 |
|---|---|---|---|---|---|---|---|
| 0.1 | 0.2 | 0.7 | **1.334** | 2.844 | 5.868 | 13.864 | 25.009 |
| 0.33 | 0.33 | 0.33 | 1.393 | 2.664 | **4.616** | **11.210** | **20.730** |
| 0.6 | 0.3 | 0.1 | 1.449 | 2.763 | 5.629 | 14.104 | 23.985 |
| **0.05** | **0.15** | **0.8** | 1.369 | **2.624** | 5.541 | 13.773 | 28.806 |

Table 6: Loss weighting coefficients for varying downsampling rates $\rho_s$.

## A.6 ATTENTION HEADS VISUALIZATIONS

The attention weights in Fig. 8 demonstrate functional specialization among attention heads. For example, Head 3 focuses on local kinematics, while Heads 2, 4, 6, and 8 act as global anchors that attend to long-term trajectory semantics. This diversity ensures a comprehensive representation that integrates fine-grained motion dynamics with high-level trip intent.

## A.7 TRAINING EFFICIENCY

We compare HiT-JEPA with baselines in terms of training time per iteration in Table 8. While TrajCL and CLEAR achieve lower training time due to their lightweight structures, HiT-JEPA remains highly competitive at rank 3 among 5 methods. Moreover, by incorporating convolution-based trajectory semantics aggregation and learning on multi-level trajectory abstractions with 1-layer Transformer backbones, HiT-JEPA remains efficient while achieving generalizable and robust performance.

## A.8 VISUALIZATIONS

We visualize two sets of comparisons of 5-NN queries after fine-tuning by Hausdorff measure in Fig. 12b and Fig. 13b, where each row shows the rank 1 to 5 matched trajectories from left to right, given red query trajectories. The rightmost figures are the indices of the query and matched

| $\lambda$ | $\mu$ | $\nu$ | 0.1 | 0.2 | 0.3 | 0.4 | 0.5 |
|---|---|---|---|---|---|---|---|
| 0.1 | 0.2 | 0.7 | 1.081 | 1.109 | 1.095 | 1.115 | 1.135 |
| 0.33 | 0.33 | 0.33 | 1.130 | 1.138 | 1.133 | 1.152 | 1.196 |
| 0.6 | 0.3 | 0.1 | 1.092 | 1.107 | 1.133 | 1.117 | 1.191 |
| **0.05** | **0.15** | **0.8** | **1.074** | **1.077** | **1.085** | **1.093** | **1.119** |

Table 7: Loss weighting coefficients for varying distortion rates $\rho_d$.

Table 8: Comparison of training efficiency (seconds per iteration).

| Method | Time (s) |
|---|---|
| TrajCL | 0.196 |
| TrjSR | 0.476 |
| CLEAR | 0.292 |
| T-JEPA | 1.022 |
| HiT-JEPA | 0.341 |

trajectories. We can find that the improvements of HiT-JEPA can find more similar trajectories on ranks 4 and 5, resulting in a higher average HR@5 than T-JEPA.

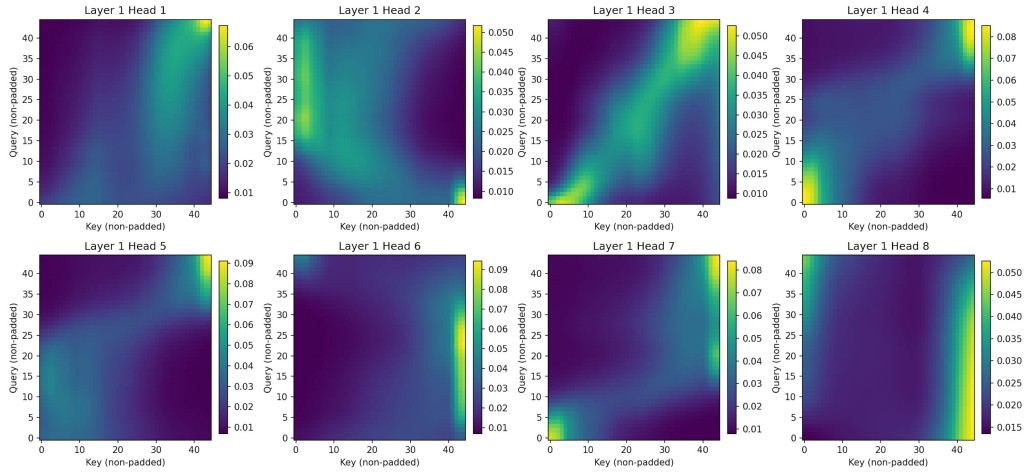

Figure 8: Visualization of each of the 8 attention heads at the JEPA layer $A^{(1)}$.

### A.9 REPRESENTATION VISUALIZATION VIA CLUSTERING

We cluster and visualize the embedding of 3000 random trajectories in Porto in Fig. 9. We use a K-Means Clusterer with a number of cluster centers $K = 6$ acquired from the Elbow Method. We can find that, although the boundaries between clusters are soft without a specific self-clustering design in recent clustering methods Yao et al. (2017); Fang et al. (2021), distinct semantic groups are clearly visually discernible. This demonstrates the strong potential of HiT-JEPA, which is trained on regression loss, to be fine-tuned to generalize to multiple trajectory tasks.

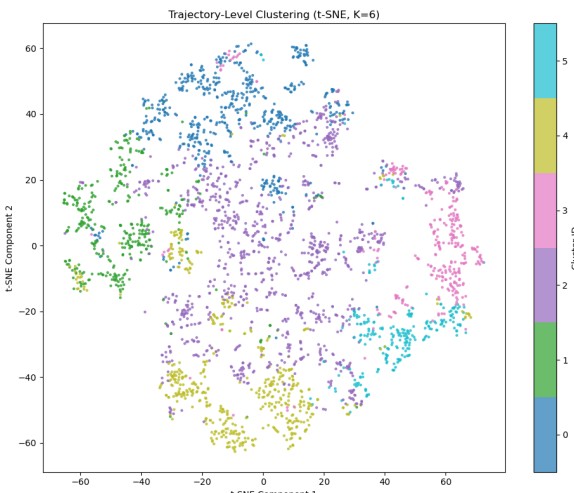

Figure 9: t-SNE Visualization of Trajectories.

## A.10 LIMITATIONS AND FUTURE WORK

By upsampling and fusing attention weights across adjacent layers, HiT-JEPA demonstrates one form of hierarchical interaction common to Transformer-based JEPA models. Therefore, one extension could be developing a unified hierarchical interaction framework for all kinds of learning architectures (e.g., CNNs, Mambas, LSTMs, etc.). This will enable each architecture to plug in its customized hierarchy module while preserving a consistent multi-level learning paradigm.

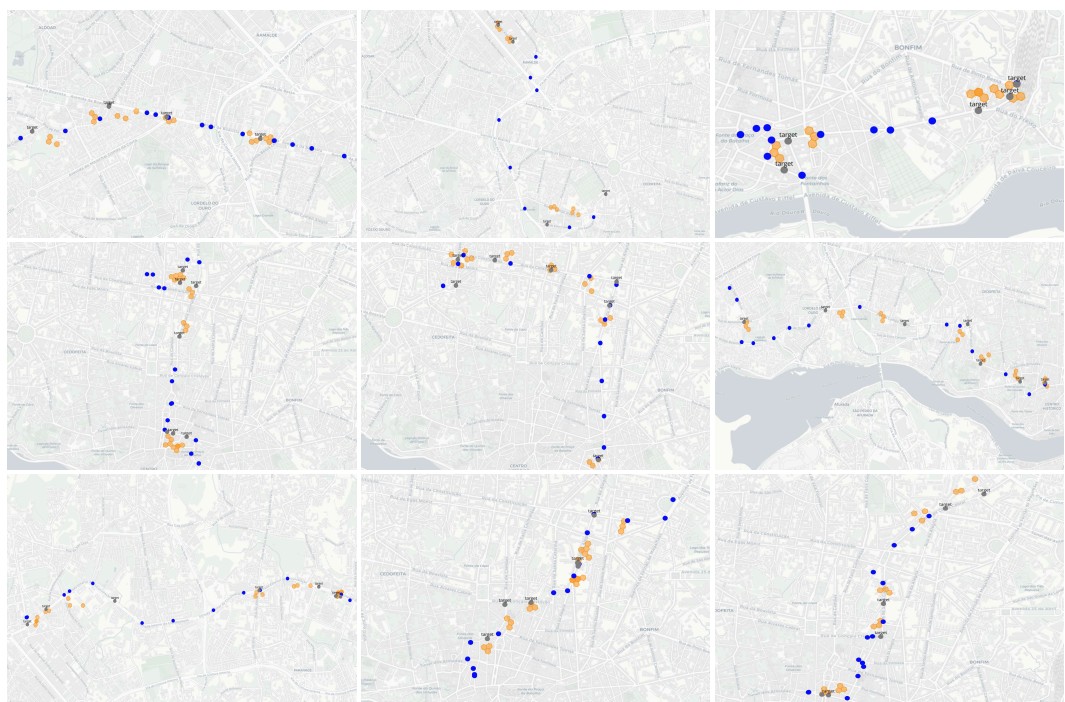

Figure 10: Visualization of predicted masked trajectories.

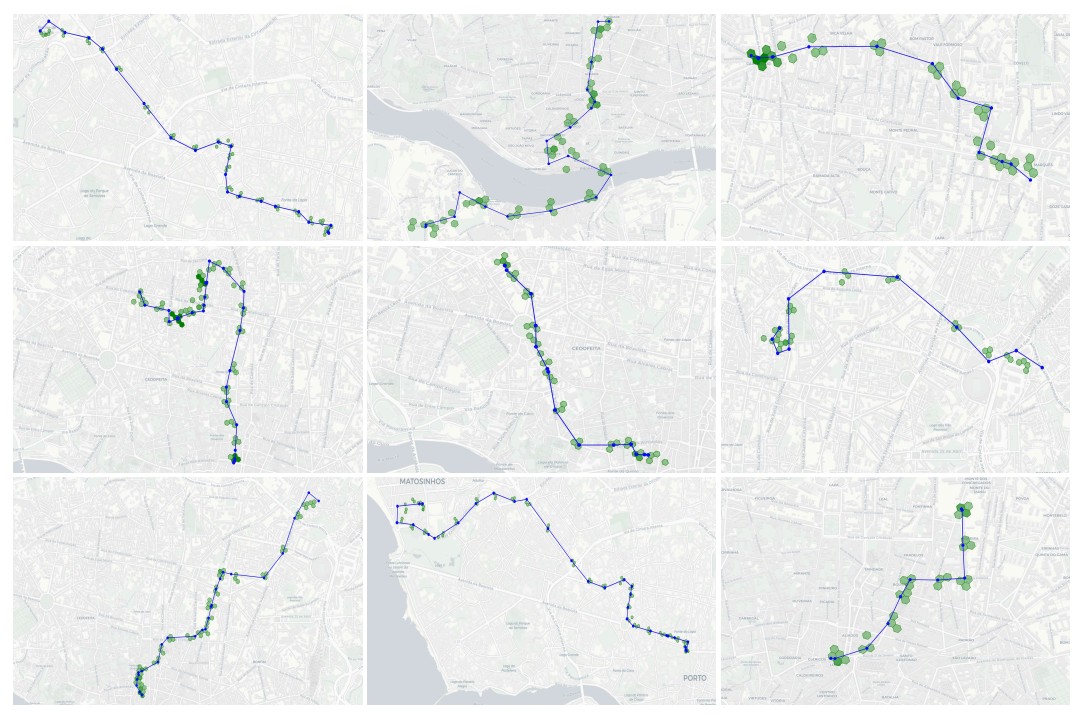

Figure 11: Visualization of encoded full trajectories.

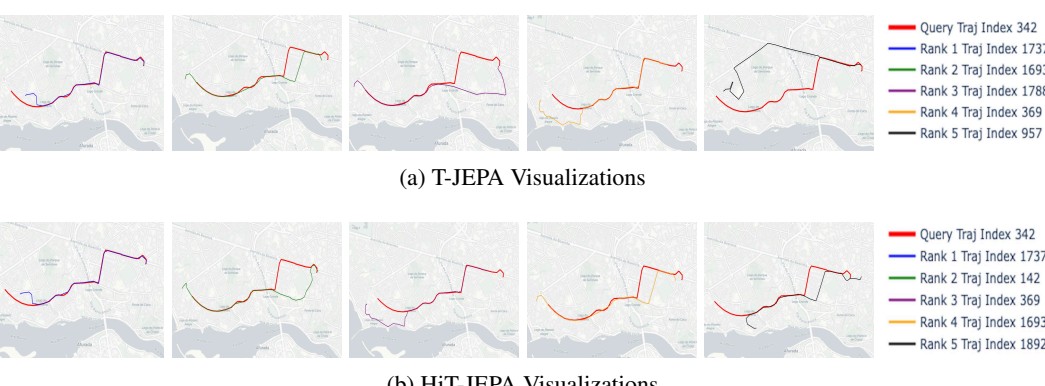

Figure 12: Comparisons of 5-NN search between T-JEPA and HiT-JEPA on Porto after being fine-tuned by Hausdorff measure.

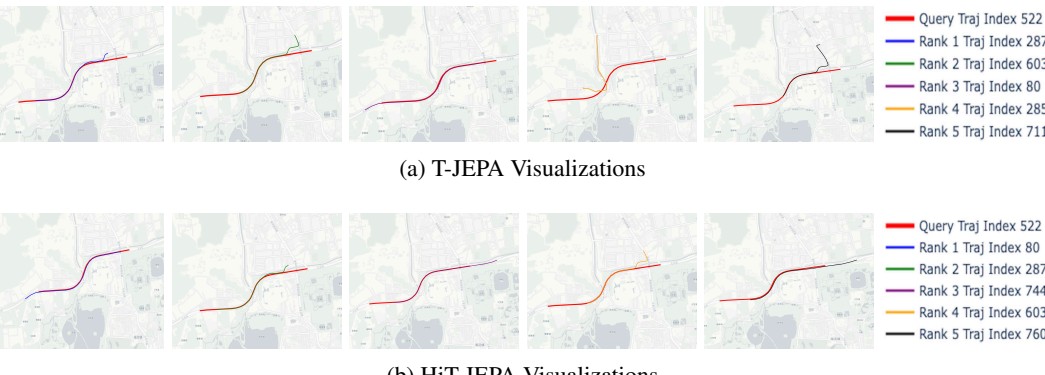

Figure 13: Comparisons of 5-NN search between T-JEPA and HiT-JEPA on GeoLife after being fine-tuned by Hausdorff measure.

