# OpenReview forum: "HiT-JEPA: A Hierarchical Self-supervised Trajectory Embedding Framework for Similarity Computation"
_ICLR.cc/2026/Conference — Submitted to ICLR 2026_

### Official Review · Reviewer_ZHZ7 · 2025-10-28

**Soundness:** 2
**Presentation:** 3
**Contribution:** 2
**Rating:** 4
**Confidence:** 2

**Summary:**

The paper proposes HiT-JEPA, a framework for learning multi-scale urban trajectory representations across different semantic abstraction levels. This framework constructs a three-layer hierarchical structure, which generates local point-level, intermediate segment-level patterns, and high-level global route structure. HiT-JEPA also includes a joint embedding predictive architecture to incoporate multiple levels of abstractions and approximate the target trajectory. Experiments on real-world urban GPS trajectory datasets show that HiT-JEPA exhibits stable performance in self-similarity search, zero-shot scenarios, and downstream fine-tuning tasks.

**Strengths:**

1. HiT-JEPA constructs a three-layer hierarchical structure within a single model using three pooling layers. It is the first architecture to explicitly unify both fine-grained and abstract trajectory patterns in one framework.

2. Experiments on multiple real-world datasets and few-shot learning tasks demonstrate its generalizability.

**Weaknesses:**

1. The paper lacks sufficient novelty and mainly appears to be an implementation of industry approaches.
2. The experimental section lacks sufficient and diverse baselines, for example transformer-based sequence forcasting methonds.
3. The paper does not provide detailed interpretations or examples to demonstrate how the learned three-layer structure captures different levels of semantics. It remains unclear whether the extracted multi-scale trajectory representations truly correspond to point-level or higher-level semantics.
4. Multi-layer Transformer struction can also capture multi-granular information: lower layers typically encode specific input semantics, intermediate layers abstract higher-level concepts, and final layers contribute to generation or prediction. It would be helpful to explain why the authors chose a multi-layer convolutional structure instead of multi-layer Transformer. Would the convolutional yield any advantage in performance or interpretability?

**Questions:**

See weaknesses.

---

> ### Author Response · Authors · 2025-11-28
> **Response to reviewer ZHZ7 (1/4)**
>
> ### **W1**: The paper lacks sufficient novelty and mainly appears to be an implementation of industry approaches.
>
>
>
> ### **A1**
> We sincerely thank the reviewer ZHZ7 for your comment. We would like to clarify that our method is not an implementation of any industry approaches. The novelty of this paper is a hierarchical JEPA structure for multi-scale urban trajectory representation learning:
>
> 1) Our motivation stems from combining the **JEPA framework** and **hierarchical self-supervised learning** to extract **multi-scale robust trajectory features**. This idea is not from any industry algorithms.
>
> 2) HiT-JEPA and other existing JEPA methods are **purely academic**, and have been proven promising by existing recent JEPA-based methods such as [I-JEPA](https://openaccess.thecvf.com/content/CVPR2023/papers/Assran_Self-Supervised_Learning_From_Images_With_a_Joint-Embedding_Predictive_Architecture_CVPR_2023_paper.pdf) and [V-JEPA](https://openreview.net/pdf?id=WFYbBOEOtv). Therefore, we compare against standard academic baselines in trajectory similarity computation rather than industrial systems, and we have also provided our code repository.
>
> 3) The motivation of HiT-JEPA is to learn multi-scale trajectory representations at different abstraction levels. We start from the characteristics of urban GPS trajectories, where they can exhibit more complicated moving patterns and sudden turnings attributed to urban layout. Existing trajectory representation learning methods model trajectories on point level, which makes the model pay attention to every change in trajectory patterns. However, some patterns may not change the overall moving pattern, such as slight distortions of certain points or moving straight along a street, but the model needs to capture this trivial information, and thus affect the embedding quality for overall trajectory features. So, the rationale of HiT-JEPA is:
>     - We extract **high-level abstractions** of trajectories to capture the **sharpest changes of patterns** in trajectories.
>     - then we **propagate** the "sharp" information into lower-level abstractions and **fuse by attention weights**.
>     - lower-level embeddings will learn **adaptive representations** by considering both "sharp" features from higher levels and "soft" features learned from the current level. We will illustrate the difference of features learned across multiple levels in our **revised manuscript**, to further strengthen this contribution.
>
>
>  We are also **the first work** that extends current JEPA frameworks to a Hierarchical JEPA (H-JEPA), where this concept was first proposed by [LeCun (2022)](https://openreview.net/pdf?id=BZ5a1r-kVsf), and no papers have developed an H-JEPA before. We further reaffirm our contributions by comparison with existing Hierarchical Self-supervised Learning (HSSL) methods:
>
> | Existing HSSLs                                                                               | **HiT-JEPA**                                                                                 |
> | -------------------------------------------------------------------------------------------- | -------------------------------------------------------------------------------------------- |
> | partition inputs as discrete fragments                                                       | creates multiple abstraction levels of the entire data sample                                |
> | directly propagate lower-level representations to higher-level encoders                      | calculate level-wise loss, then hierarchical interactions by attention weights gated fusion |
> | single loss on outputs from the highest level                                                | weighted losses across all levels                                                            |

---

> ### Author Response · Authors · 2025-11-28
> **Response to reviewer ZHZ7 (2/4)**
>
> ### **w2**: More baseline comparisons.
>
>
> ### **A2**
>
> We apply our experimental settings on one more baseline **TrjSR** by Cao *et al.* (2021), which is a generative model that converts trajectories into images and then adopts image super-resolution to learn their patterns. We will also add this baseline with reference in our revised manuscript.
>
> Here we compare the diversity of our baselines:
>
> | Method   | Backbones                | Learning Frameworks            |
> |----------|--------------------------|--------------------------------|
> | TrajCL   | Dual-feature Transformer |       Contrastive Learning     |
> | TrjSR    |     CNNs                 | Single Image Super-resolution  |
> | CLEAR    |     Gated Recurrent Unit |     Contrastive Learning       |
> | T-JEPA   |       Transformer        |             JEPA               |
>
>
> And we compare TrjSR with HiT-JEPA on all experiments:
>
> 1. Self-similarity:
>
> | Dataset | Method | R1 $\vert D\vert$ | R1 $\rho_{s}$ | R1 $\rho_{d}$ | R2 $\vert D\vert$ | R2 $\rho_{s}$ | R2 $\rho_{d}$ | R3 $\vert D\vert$ | R3 $\rho_{s}$ | R3 $\rho_{d}$ | R4 $\vert D\vert$ | R4 $\rho_{s}$ | R4 $\rho_{d}$ | R5 $\vert D\vert$ | R5 $\rho_{s}$ | R5 $\rho_{d}$ |
> | :--- | :--- | :--- | :--- | :--- | :--- | :--- | :--- | :--- | :--- | :--- | :--- | :--- | :--- | :--- | :--- | :--- |
> | **Porto** | TrjSR | 3.240 | 12.553 | 12.509 | 5.321 | 16.945 | 15.401 | 7.073 | 37.150 | 15.901 | 8.740 | 65.413 | 28.914 | 10.192 | 149.950 | 32.730 |
> | | HiT-JEPA | **1.027** | **1.339** | **1.077** | **1.046** | **2.318** | **1.081** | **1.049** | **4.440** | **1.091** | **1.059** | **11.961** | **1.099** | **1.069** | **28.770** | **1.107** |
> | **T-Drive** | TrjSR | 110.726 | 674.16 | 581.776 | 223.841 | 795.331 | 572.944 | 356.941 | 870.73 | 566.816 | 475.872 | 960.404 | 545.278 | 592.146 | 1033.404 | 566.696 |
> | | HiT-JEPA | **1.040** | **1.056** | **1.035** | **1.040** | **1.079** | **1.031** | **1.040** | **1.131** | **1.035** | **1.041** | **1.302** | **1.038** | **1.041** | **2.182** | **1.031** |
> | **GeoLife** | TrjSR | 6.765 | 8.332 | 7.747 | 7.393 | 8.594 | 7.942 | 7.661 | 8.688 | 7.648 | 7.767 | 8.566 | 8.534 | 8.350 | 8.770 | 9.460 |
> | | HiT-JEPA | **1.033** | **1.058** | **1.085** | **1.033** | **1.089** | **1.211** | **1.033** | **1.171** | **1.136** | **1.034** | **1.210** | **1.202** | **1.034** | **1.403** | **1.294** |
> | **TKY** (zero-shot) | TrjSR | 8.673 | 31.770 | 27.505 | 17.120 | 37.070 | 30.758 | 22.310 | 48.985 | 30.923 | 26.820 | 64.380 | 33.113 | 29.318 | 84.302 | 34.043 |
> | | HiT-JEPA | **1.515** | **2.175** | **1.947** | **1.625** | **2.848** | **1.983** | **1.738** | **5.920** | **1.950** | **1.847** | **12.317** | **1.973** | **1.955** | **16.453** | **1.997** |
> | **NYC** (zero-shot) | TrjSR | 3.929 | 5.457 | 6.307 | 4.793 | 5.171 | 7.950 | 5.457 | 8.350 | 6.679 | 5.821 | 12.757 | 7.443 | 6.007 | 14.329 | 7.907 |
> | | HiT-JEPA | **1.393** | **1.857** | **1.571** | **1.414** | **2.400** | **1.536** | **1.450** | **1.679** | **1.543** | **1.514** | **2.571** | **1.593** | **1.564** | **4.557** | **1.543** |
> | **AIS(AU)** (zero-shot) | TrjSR | 692.000 | 3658.400 | 3649.450 | 1390.364 | 3661.421 | 3649.407 | 2136.271 | 3675.043 | 3649.150 | 2942.586 | 3714.564 | 3649.086 | 2892.264 | 3700.221 | 3649.371 |
> | | HiT-JEPA | **1.336** | **3.932** | **2.478** | **1.739** | **6.991** | **2.474** | **2.051** | **11.135** | **2.474** | **2.313** | **18.058** | **2.466** | **2.475** | **24.070** | **2.474** |
>
>
> 2. Approximating heuristic measures:
>
> | Dataset | Method | EDR HR@5 | EDR HR@20 | EDR R5@20 | LCSS HR@5 | LCSS HR@20 | LCSS R5@20 | Haus HR@5 | Haus HR@20 | Haus R5@20 | Fre HR@5 | Fre HR@20 | Fre R5@20 | Average |
> | :--- | :--- | :--- | :--- | :--- | :--- | :--- | :--- | :--- | :--- | :--- | :--- | :--- | :--- | :--- |
> | **Porto** | TrjSR | 0.085 | 0.083 | 0.157 | 0.162 | 0.197 | 0.292 | 0.166 | 0.192 | 0.304 | 0.157 | 0.173 | 0.288 | 0.188 |
> | | HiT-JEPA | **0.157** | **0.195** | **0.337** | **0.367** | **0.554** | **0.717** | **0.457** | **0.584** | **0.816** | **0.403** | **0.545** | **0.752** | **0.490** |
> | **T-Drive** | TrjSR | 0.076 | 0.068 | 0.114 | 0.076 | 0.080 | 0.118 | 0.095 | 0.090 | 0.143 | 0.098 | 0.094 | 0.145 | 0.100 |
> | | HiT-JEPA | **0.095** | **0.166** | **0.246** | **0.219** | **0.379** | **0.487** | **0.191** | **0.282** | **0.401** | **0.142** | **0.201** | **0.298** | **0.258** |
> | **GeoLife** | TrjSR | 0.138 | 0.246 | 0.443 | 0.229 | 0.330 | 0.479 | 0.492 | 0.439 | 0.692 | 0.383 | 0.362 | 0.614 | 0.404 |
> | | HiT-JEPA | **0.189** | **0.415** | **0.564** | **0.253** | **0.522** | **0.609** | **0.603** | **0.697** | **0.854** | **0.492** | **0.552** | **0.834** | **0.549** |
>
>
> We can see that TrjSR is less robust to low and irregularly-sampled trajectory datasets like T-Drive, and also has limited generalization on zero-shot setting to new cities and on approximating to heuristic measures.

---

> ### Author Response · Authors · 2025-11-28
> **Response to reviewer ZHZ7 (3/4)**
>
> ### **w4**: Why multi-layer Convolutions instead of multi-layer Transformers.
>
> ### **A4**
>
> We appreciate the reviewer's insightful comment for further explanation of the convolution-based multi-level abstraction creation. We list several reasons that we deliberately design the hierarchical structure in convolutions instead of transformers:
>
> 1. **Hierarchical tokenization:**
>     - In trajectory data, single GPS points are considered "characters" in NLP or "pixels" in Computer Vision, which are noisy and usually carry low-level and low-density information. Therefore, we apply **convolutional layers** as learnable tokenizers to aggregate the low-level "characters" locally to higher-level "words", then to "sentences". And feed the hierarchical tokens into the Transformer encoders to capture the dependencies among the semantic tokens at that level.
>     - The convolutional kernels aggregate local points into more robust and meaningful tokens. For example, the aggregated pattern can describe that several points "go straight" even though there are mild distortions, or aggregated points in an intersection into one token that exhibits a "turning" action.
>     - **Transformers**, otherwise capture the dependencies among tokens at that specific level. Existing classical applications of Transformers like BERT, Vision-Transformer, and their variants all rely on a specified level of tokens: language words or image patches as meaningful tokens, then apply the transformer encoder to extract the long-term dependencies of the tokens. Therefore, the multi-level tokens determine what levels of information for the Transformer encoder to learn, and prevent the Transformer to learn repetitive information from only one-level input.
>
> 2. **Local + Global inductive bias:**
>     - The locality and strong inductive bias of **CNNs** help with aggregating local trajectory dynamics, while the global and weak inductive bias of **Transformers** establish connections for all input tokens and calculate how much attention should be paid to each pair of tokens.
>     - By combining both, we construct higher-level tokens by **CNNs** and learn their dependencies by **Transformers**, followed by cross-level attention weight fusion and level-wise learning objectives, to learn trajectory dynamics in multiple semantic levels.
>     - We will upload a **revised manuscript** with interpretations of multi-level attention weights to explain the features learned at each level.
>
>
> 3. **Computational Efficiency**
>     - Given a trajectory with length n, **CNN** has time complexity of *O(n)* while **Vanilla Transformer** has *O(n^2)*. Therefore, CNN is an efficient and linear scale approach to construct the multi-level tokens.

---

> ### Author Response · Authors · 2025-12-03
> **Response to reviewer ZHZ7 (4/4)**
>
> ### **W3**: Interpretation of hierarchical design.
>
> ### **A3**
>
> We have visualized and explained in detail the interpretability of hierarchical attention weights, and added them in Section 4.3 and Appendices A.6 in our revised manuscript. We give a case study of the model attention at each JEPA level corresponding to a physical trajectory. We provide the raw attention map at each level (with each of the level 1 attention heads visualizations), the attention deviation heatmap, and attention statistical profiles to jointly analyze the trajectory semantics learned across all levels.

---

### Official Review · Reviewer_19oU · 2025-10-31

**Soundness:** 3
**Presentation:** 3
**Contribution:** 2
**Rating:** 6
**Confidence:** 4

**Summary:**

The paper presents HiT-JEPA, a hierarchical self-supervised framework for trajectory representation that explicitly models point-level, segment-level, and trajectory-level semantics with cross-level interactions. A JEPA (Joint Embedding Predictive Architecture) objective aligns representations in embedding space, complemented by VICReg-style regularization to prevent collapse. Experiments across multiple cities/modalities (including AIS) show strong retrieval performance and promising zero-shot generalization.

**Strengths:**

- Address single-scale bias by introducing an explicit multi-scale hierarchy with information flow across levels.
- Propose a JEPA-style training reduces reliance on heavy data augmentation; ablations indicate the benefit of hierarchical interactions.
- Extensive experiments across datasets and zero-shot settings supports generalization claims.

**Weaknesses:**

- Only two contrastive learning baselines are compared in the experiments. There are many trajectory representation learning works should be compared.
- The interpretations of the results of HiT-JEPA are quite confusing. It would be better to explain why the two showcases can illustrate the interpretation of the proposed method.
- For different trajectory abstraction method, the results would change. From equation (3)-(5), the sampling rate are 50% and 25% of orginal trajectories which may not be the best setting.  How  the abstraction method influence the results should be disscussed.
- The efficiency of JEPA should be considered.

**Questions:**

See the weakness above.

---

> ### Author Response · Authors · 2025-11-30
> **Response to reviewer 19oU (1/4)**
>
> ### **w1**: Compare more baselines in the experiments.
>
> ### **A1**
>
> We sincerely thank reviewer 19oU for your constructive comments.
>
> We conduct our experiments on one more baseline **TrjSR** by Cao *et al.* (2021), the comparisons can be referred to our **Response to reviewer ZHZ7 (2/4)**. We would like to restate our baseline selection strategy and summarize it in the table below:
>
>
> | Method   | Backbones                | Learning Frameworks            |
> |----------|--------------------------|--------------------------------|
> | TrajCL   | Dual-feature Transformer |       Contrastive Learning     |
> | TrjSR    |     CNNs                 | Single Image Super-resolution  |
> | CLEAR    |     Gated Recurrent Unit |     Contrastive Learning       |
> | T-JEPA   |       Transformer        |             JEPA               |
>
> Therefore, we select baselines on diverse backbones and learning frameworks. Since the primary focus of this work is trajectory similarity computation, we only compare with similarity methods.

---

> ### Author Response · Authors · 2025-12-01
> **Response to reviewer 19oU (2/4)**
>
> ### **W4**: Efficiency comparisons.
>
> ### **A4**
>
> We thank the reviewer for raising the question regarding efficiency. We compare HiT-JEPA with baselines on training time for each training iteration in the table below. During the rebuttal phase, we have conducted a comprehensive code optimization. Therefore, we report the optimized efficiency:
>
> | Method | Time (s) |
> | :--- | :--- |
> | TrajCL | 0.196 |
> | TrjSR | 0.476 |
> | CLEAR | 0.292 |
> | T-JEPA | 1.022 |
> | HiT-JEPA | 0.341 |
>
> We can find that while TrajCL and CLEAR achieve lower training time due to their lightweight structures, HiT-JEPA remains highly competitive at rank 3 among 5 methods. Moreover, by incorporating convolution-based trajectory semantics aggregation and learning on multi-level trajectory abstractions with 1-layer Transformer backbones, HiT-JEPA remains efficient while achieving generalizable and robust performance. We have updated this comparison in Appendices A.7 in our revised manuscript.

---

> ### Author Response · Authors · 2025-12-02
> **Response to reviewer 19oU (3/4)**
>
> ### **W3**: Influence of different downsampling rates.
>
> ### **A3**
>
> We thank the reviewer 19oU for this insightful question regarding our trajectory abstraction design. We clarify that the **kernel size of convolutional layers is 3** and the downsampling is achieved by **a stride of 2**. We have updated this detail in Section 3 of our revised manuscript. The reasons for the downsampling operations of 50% and 25% can be concluded as:
>
> 1. **Smooth semantic granularity transition:**
>     - Physically, aggregating two points into one feature serves as a natural abstraction from absolute positions to local displacements (N -> N/2), then to summarized moving behaviors (N/2 -> N/4). Therefore, the 50% and 25% downsampling operations provide a smooth and logical transition between the granularities, ensuring a continuous feature hierarchy.
>     - With a kernel size of 3, each aggregated point also incorporates context from its near neighbors and is critical for preserving the temporal continuity of the trajectory, preventing information loss associated with aggressive downsampling.
>
> 2. **Feature Alignment during Upsampling:** our hierarchical design requires precise feature alignment between levels.
>     - For the upsampling process in Eq. 13, using an odd stride like 3 will cause asymmetric padding/cropping during the upsampling process. This introduces boundary artifacts and accumulated feature alignment in deeper JEPA levels.
>     - Using a stride of 2 ensures symmetric upsampling, which maintains feature alignment and is computationally more efficient.
>
> 3. **Robustness for short trajectories:** we process, train, and test on trajectories with lengths ranging from [20, 200].
>     - Consider an extreme scenario with a trajectory of **length 20**.
>     - Assuming that we apply a **stride of 3**, the length at each level will rapidly degrade from 20 to 6, then to 2. A sequence of 2 is insufficient for the Transformer to model any meaningful global movement.
>     - If using a **stride of 2**, same as in our manuscript, there are still 5 points at level 3 for the model to capture valid temporal dependencies.
>
> **Additional optimization**: At the same time, we have observed that maintaining a consistent embedding dimension across higher levels, rather than increasing it, can reduce time and space complexity while achieving slight performance gains. We have updated equations **Eq. 4 and Eq.5** with a consistent embedding dimension $d=256$ along with the experimental results in the revised manuscript.

---

> ### Author Response · Authors · 2025-12-03
> **Response to reviewer 19oU (4/4)**
>
> ### **W2**: Interpretation of hierarchical design.
>
> ### **A2**
>
>
> We have visualized and explained in detail the interpretability of hierarchical attention weights, and added them in Section 4.3 and Appendices A.6 in our revised manuscript. We give a case study of the model attention at each JEPA level corresponding to a physical trajectory. We provide the raw attention map at each level (with each of the level 1 attention heads visualizations), the attention deviation heatmap, and attention statistical profiles to jointly analyze the trajectory semantics learned across all levels.

---

### Official Review · Reviewer_uZkJ · 2025-11-01

**Soundness:** 2
**Presentation:** 3
**Contribution:** 2
**Rating:** 2
**Confidence:** 5

**Summary:**

This paper introduces HiT-JEPA, a hierarchical self-supervised framework for learning urban trajectory embeddings suitable for similarity computation. HiT-JEPA employs a three-level joint embedding predictive architecture (JEPA) to progressively capture point-level, segment-level, and global trajectory abstractions, integrating top-down and multi-scale semantic information. The approach is evaluated extensively against state-of-the-art baselines on multiple urban trajectory datasets, including zero-shot, and downstream tasks. Results claim remarkably improvements in mean ranking performance, generalization, and ablation integrity on various trajectory similarity and retrieval tasks.

**Strengths:**

S1. HiT-JEPA proposes an explicit three-layer architecture that learns and aligns representations at point, segment, and trajectory levels.

S2. The method implements a novel attention propagation strategy between abstraction levels. And it upsamples higher-level attention maps to guide feature extraction at lower levels.

S3. HiT-JEPA demonstrates exceptional zero-shot performance across heterogeneous datasets. Experimental design is rigorous and covers critical scenarios like self-similarity, ablation and visualization.

**Weaknesses:**

W1. This manuscript has presented limited novelty comparing to prior works. The JEPA and multi-scale structure build heavily on established elements in self-supervised learning, attention propagation, and urban trajectory modeling. The direct design leap over T-JEPA, is incremental rather than a decisive paradigm shift, especially given already cited work such as HIBERT and HiCLRE in NLP and CV. So, this work simply applies these models to the trajectory dataset without making more targeted designs based on the characteristics of trajectory data.

W2. The manuscript claims high-level attention "guides local feature extraction" but does not clarify what specific semantic information high-level attention captures. The authors should provide specific explanations regarding the "semantic information," such as the clustering characteristics of trajectories or the motion features of moving objects. They may even need to include a case study to illustrate this point. Otherwise, the high-level attention mechanism remains a black box, merely effective by coincidence in the current dataset.

W3. HiT-JEPA uses Uber H3 grids to map GPS coordinates instead of traditional rectangular grids but failing to explain how H3’s hexagonal symmetry improves trajectory spatial relation modeling. The effectiveness of these two grid-based methods should also be compared experimentally.

**Questions:**

Q1. Provide more description about the trajectory-specific model designs, explaining how these designs enhance the effectiveness of trajectory similarity computation, and discuss the differences when applied in the NLP and CV domains.

Q2. Add attention-weight visualizations and controlled experiments to clarify and show how it quantifies the guidance. Or add a case study to show the effectiveness of a high-level attention mechanism.

Q3. Conduct an ablation study comparing H3 grids with rectangular grids on selected datasets.

---

> ### Author Response · Authors · 2025-12-01
> **Response to reviewer uZkJ (1/3)**
>
> ### **W3** and **Q3**: Comparisons between h3 cells and rectangular cells.
>
> ### **A3**
>
> We sincerely thank the reviewer uZkJ for your dedicated comments. Since existing free-space trajectory modelling methods discretize the spatial regions into cells and pre-train the structural relations between them, the consistency of connectivity between one cell and its neighbors is very important to learn more accurate spatial relations:
>
> - **Anisotropy of rectangular cells versus Isotropy of H3 hexagonal cells**:
>     - **Rectangular cells have Anisotropy**. For the center of a rectangular cell, the distances from the centers of axial neighbors and the centers of diagonal neighbors are different, which are $1x$ and $\sqrt{2}x$ given the side lengths $x$. When we tokenize the trajectory points, those that fall into the diagonal and axial neighbors of the same rectangular cell are treated with identical connectivity, ignoring the physical distance discrepancy.
>     - **Hexagon cells have Isotropy**, where each of them has 6 axial neighbors with equal center-to-center distance. This is crucial for capturing accurate physical dynamics.
> - **Distortions near the Poles**:
>     - Rectangular grids incur increasing projective distortion near the poles due to spherical topology
>     - The global consistency provided by H3 prevents the necessity to deal with the artifacts caused by projection distortion.
> - H3 hexagonal cells have been used for some existing urban trajectory modelling methods [Musleh *et al.*, 2023](https://par.nsf.gov/servlets/purl/10538601) and [Faraji *et al.*, 2023](https://dl.acm.org/doi/pdf/10.1145/3589132.3625619), which validates the effectiveness of hexagonal indexing for robust spatial representation.
>
>
> We further provide the experimental results of HiT-JEPA self-similarity search on the Porto dataset between use rectangular cells and hexagonal cells:
>
> **For varying DB sizes $|\mathcal{D}|$**
>
> | Method                | R1      | R2      | R3      | R4      | R5      |
> |-----------------------|---------|---------|---------|---------|---------|
> | HiT-JEPA (rectangular)| 1.032  | 1.062  | 1.069  | 1.080  | 1.093 |
> | HiT-JEPA (**H3**)         | **1.026** | **1.043**  | **1.048**   | **1.058**   | **1.065**   |
>
>
> **For varying downsampling rates $\rho_{s}$:**
>
> | Method                | R1      | R2      | R3      | R4      | R5      |
> |-----------------------|---------|---------|---------|---------|---------|
> | HiT-JEPA (rectangular)| 1.508  | 3.054  | 7.735  | 18.912  | 36.768 |
> | HiT-JEPA (**H3**)         | **1.369** | **2.624**  | **5.541**   | **13.773**   | **28.806**  |
>
>
> **For varying distortion rates $\rho_{d}$:**
>
> | Method                | R1      | R2      | R3      | R4      | R5      |
> |-----------------------|---------|---------|---------|---------|---------|
> | HiT-JEPA (rectangular)| 1.095  | 1.111  | 1.123  | 1.122  | 1.124 |
> | HiT-JEPA (**H3**)         | **1.074** | **1.077**   | **1.085**   | **1.093**   | **1.119**   |
>
> In the process of conducting additional experiments, we performed minor refinements of the hyperparameters and yielded performance gains on most of the metrics compared to the original submission. We will **update the parameters and experimental results in our revised manuscript**. Here, we report the comparison between rectangular grids and H3 hexagonal grids on refined hyperparameters.

---

> ### Author Response · Authors · 2025-12-02
> **Response to reviewer uZkJ (2/3) (Part 1 of 2)**
>
> ### **W1 and Q1**: Explanation of trajectory-specific design and their effectiveness, and novelty compared to existing HSSL methods in CV and NLP.
>
> ### **A1**
>
> We thank the reviewer uZkJ for raising the question regarding the novelty of HiT-JEPA.
>
> 1. **Trajectory-specific hierarchical tokenizations**.
>     - **NLP/CV**: in NLP, inputs are discrete tokens **pre-defined semantics** like natural language words, where each token has their own meaning and is ready to interact with others. In CV, e.g., ViT, inputs are static pixel patches which contain **fixed local structural information**.
>     - **Trajectory gap**: a single GPS point has very limited semantic meaning on its own. Meaning arises only from the continuous kinematic sequence (velocity, acceleration, turns). Unlike text, trajectory data has continuous, irregularly-sampled, and noisy characteristics without natural "word" boundaries. Existing methods often treat points as isolated token IDs (mostly rectangular cells) and learn the token dependencies by Transformer-based or RNN-based backbones. However, existing approaches typically treat points as isolated IDs, with less attention paid to inherent kinematic continuity.
>     - **Our novelty**: we adopt simple but effective convolutional layers (with kernel size of 3 and stride of 2, updated in Section 3) to create multi-level trajectory semantics. With a kernel size of 3, each trajectory point at each JEPA level incorporates neighboring context to construct cohesive local features. And a stride of 2 extracts trajectory kinematics from absolute positions to local displacements (N -> N/2) from level 1 to level 2, then to more summarized moving behaviors (N/2 -> N/4) from level 2 to level 3. By this approach, we ensure that higher levels can describe moving behaviors by convolved tokens without losing temporal continuity. More details regarding such design rationale can be referred to our **response to reviewer 19oU (3/4)**.
>
> We support the effectiveness of our trajectory-specific hierarchical tokenizations by comparing HiT-JEPA **without** convolutional-based abstraction method, and replacing it with max pooling layers with a stride of 2 only to achieve the same downsampling rate:
>
> **For varying DB sizes $|\mathcal{D}|$**
>
> | Method                | R1      | R2      | R3      | R4      | R5      |
> |-----------------------|---------|---------|---------|---------|---------|
> | HiT-JEPA (no_token)| 1.051 | 1.104 | 1.122 | 1.157 | 1.204 |
> | HiT-JEPA          | **1.026** | **1.043**  | **1.048**   | **1.058**   | **1.065**   |
>
>
> **For varying downsampling rates $\rho_{s}$:**
>
> | Method                | R1      | R2      | R3      | R4      | R5      |
> |-----------------------|---------|---------|---------|---------|---------|
> | HiT-JEPA (no_token)| 2.353 | 7.567 | 21.175 | 59.692 | 110.471 |
> | HiT-JEPA         | **1.369** | **2.624**  | **5.541**   | **13.773**   | **28.806**  |
>
>
> **For varying distortion rates $\rho_{d}$:**
>
> | Method                | R1      | R2      | R3      | R4      | R5      |
> |-----------------------|---------|---------|---------|---------|---------|
> | HiT-JEPA (no_token)| 1.251 | 1.279 | 1.326 | 1.356 | 1.552 |
> | HiT-JEPA         | **1.074** | **1.077**   | **1.085**   | **1.093**   | **1.119**   |
>
> We can find that, without the convolutional layers to aggregate higher-level semantic behaviors, the model suffers from learning accurate and robust embeddings, especially for the varying downsampling experiment.

---

> ### Author Response · Authors · 2025-12-02
> **Response to reviewer uZkJ (2/3) (Part 2 of 2)**
>
> 2. **Explicit level-wise feature extraction via Hierarchical JEPA vs. Implicit feature Hierarchy**.
>     - **For NLP/CV**: Existing HSSL methods rely on **natural boundaries** found in data. In NLP (e.g., HIBERT) specify its hierarchies into word, sentence, and document levels, which are based on the natural semantics of human languages. In CV (e,g, *Chen et al. 2022* cited in Section 2), images are manually defined by cell, patch, and region levels. These methods simply aggregate features from lower layers, which works well for data with static or syntactic boundaries.
>     - **Our novelty**: in contrast to natural languages and images, trajectory exhibits **more continuous patterns by kinematics** connecting discrete points. Using a naive patch-based method to extract hierarchical trajectory features might disconnect the kinematic or behavioral information between trajectory patches.
>         - **Hierarchical trajectory abstractions (hierarchical token construction)**: we use convolutional layers to extract higher-level trajectory semantics while preserving continuous relations, enhancing level-specific trajectory semantic learning.
>         - **Explicit vs. Implicit**: Compared to existing HSSL methods, where higher levels directly receive lower-level **implicit** output, HiT-JEPA learn **explicit** level-specific trajectory features by individual JEPAs. With level-wise loss calculation, each JEPA level can learn distinct semantic and behavioral patterns. Benefited by our proposed hierarchical interaction module, HiT-JEPA adaptively fuses multi-scale trajectory semantics to yield more comprehensive embeddings. We have **updated the visualizations of attention weights** at each JEPA level in Section 4.2 in our revised manuscript, which strengthens our claim on learning distinct and interpretable features at each level.
>
>
>
> We summarize our key novelties compared to existing HSSL methods.
>
>
> | Existing HSSLs                                                                               | **HiT-JEPA**                                                                                 |
> | -------------------------------------------------------------------------------------------- | -------------------------------------------------------------------------------------------- |
> | partition input based on pre-defined natural boundaries, treating tokens as static discrete fragments.     | creates multiple abstraction levels of the entire trajectory, summarizing local dynamics into higher-level behavioral tokens                  |
> | Implicitly propagate lower-level representations to higher-level encoders                      | learns explicit hierarchical interactions by attention weights gated fusion, ensuring cross-level semantic consistency. |
> | single loss on outputs from the highest level                                                | weighted losses across all levels, forcing each layer to learn specific semantic features         |
>
>
> Therefore, this work proposes a **paradigm shift in how trajectory semantics are abstracted (tokenized) and hierarchically encoded**, providing a complete framework validated by similarity search experiments.

---

> ### Author Response · Authors · 2025-12-03
> **Response to reviewer uZkJ (3/3)**
>
> ### **W2 and Q2**: Add attention weights to hierarchical design and multi-level semantics.
>
> ### **A2**
>
>
> We have visualized and explained in detail the interpretability of hierarchical attention weights, and added them in Section 4.3 and Appendices A.6 in our revised manuscript. We give a case study of the model attention at each JEPA level corresponding to a physical trajectory. We provide the raw attention map at each level (with each of the level 1 attention heads visualizations), the attention deviation heatmap, and attention statistical profiles to jointly analyze the trajectory semantics learned across all levels.

---

### Official Review · Reviewer_db7G · 2025-11-16

**Soundness:** 3
**Presentation:** 3
**Contribution:** 3
**Rating:** 4
**Confidence:** 4

**Summary:**

This paper introduces a new trajectory representation approach that models trajectory semantics through three hierarchical layers at different granularities: point, segment, and the whole trajectory. Specifically, it follows similar ideas to JEPA and T-JEPA by training a trajectory encoder for each layer and applying different random masks to encourage the model to predict the masked portions. Experiments on several benchmarks for trajectory similarity search tasks demonstrate the generalization and effectiveness of the proposed method.

**Strengths:**

S1. The paper is easy to read, and the argument that existing methods lack hierarchical representations is both interesting and important.
S2. The methodology, although based on techniques used in previous studies, is sound and well-justified.

**Weaknesses:**

W1. Although it is interesting to consider different levels of trajectory semantics, the authors do not provide sufficient experimental evidence to show the effectiveness of their approach. For example, it is unclear which representation layers (S1, S2, S3, or their combination) are used for the trajectory similarity search experiments. These details appear to be missing. If the proposed framework indeed works as claimed, the different representations should capture distinct semantic meanings of trajectories. Therefore, different tasks tailored to each learned representation should be included. For example, trajectory clustering, which groups trajectories based on high-level semantic similarities, should be compared with existing deep-learning-based clustering methods. Additionally, behavioral pattern detection would be an ideal task to demonstrate the usefulness of intermediate representations. Currently, the experiments are limited to trajectory similarity tasks (self-similarity and similarity search via fine-tuning), which are insufficient to demonstrate the advantages of hierarchical representations.

W2. The use of self-similarity does not very meaningful in this context. Since HiT-JEPA is trained with random masking of trajectory points, strong self-similarity results are somewhat expected. The paper focuses heavily on similarity matching as the sole measure of embedding quality, which is inadequate. Moreover, for the Porto dataset, where performance is worse than on others, the authors attribute this to higher sampling frequency. While this explanation is plausible, the paper should more explicitly discuss this limitation and clarify the scenarios where the proposed method may or may not be effective.

W3. Can the learned encoder be applied to trajectories from different geographic regions without retraining from scratch? Discussing this aspect would strengthen the claims about generalization and practical applicability.

**Questions:**

Please see above.

---

> ### Author Response · Authors · 2025-11-29
> **Response to reviewer db7G (1/3)**
>
> ### **w3**: Transferrability to different geographic regions.
>
> ### **A3**
> We sincerely thank reviewer db7G for your insightful comments.
>
> 1. The learned encoder of HiT-JEPA can definitely be adapted to different cities. **Table 1** presents the **zero-shot** self-similarity search on TKY, NYC, and AIS(AU) using a HiT-JEPA model pre-trained on Porto with no further fine-tuning. The overall best results confirm that:
>
>     - HiT-JEPA can **accurately** capture the spatiotemporal features of trajectories, which proves that the model is learning universal representations rather than fitting to specific coordinate ranges.
>     - HiT-JEPA can not only learn good representations from vehicle GPS routes, but also learn strong **human mobility patterns** from check-in sequences (TKY and NYC).
>     - HiT-JEPA can be extended to learn *non-urban trajectories** in much **broader geographical ranges** (AIS(AU)).
>
> 2. The fine-tuning experiments conducted in **Table 2** with the highest averaged hit ratios demonstrate generalization of HiT-JEPA on multiple heuristic measures, which further validates the versatility and robustness of HiT-JEPA embeddings. By fine-tuning with a simple 2-layer MLP, we successfully generalize HiT-JEPA to have each of the following characteristics:
>     - **global spatial** sensitivity from **Hausdorff** which measures the greatest deviation between two paths,
>     - **topological continuity** from **Discrete Fréchet**, which measures the sequential topology and flow of two trajectories,
>     - **robustness to noise** from **LCSS**, which measures the longest common sub-sequence ignoring the outliers,
>     - and **invariant to low and irregular-sampling rates** from **EDR**, which measure the cost of transforming one to another.
>
> Therefore, HiT-JEPA can not only generalize across different geographical regions but also across different types of trajectories, and it has versatile embedding spaces.

---

> ### Author Response · Authors · 2025-12-02
> **Response to reviewer db7G (2/3)**
>
> ### **W1**: The workflow of HiT-JEPA, and experimental evidence of the hierarchical design.
>
> ### **A1**
>
> We thank reviewer db7G for raising the question regarding the effectiveness of HiT-JEPA design.
>
> **To answer your question for HiT-JEPA workflow directly: for the similarity search tasks, we utilize the output embedding $S'^{(1)}$ from the context encoder at Level 1. Crucially, this is not a raw low-level feature; thanks to our hierarchical interaction mechanism, $S'^{(1)}$ is a unified representation that has already integrated high-level semantic guidance (from Levels 2 and 3) via top-down attention fusion.**
>
> 1. **The detailed workflow of HiT-JEPA**: during training, we calculate the loss between the predicted representation $\widetilde{S}{'}^{(l)}(i)$ and encoded target representations $S^{(l)}(i)$ according to Eq. 15. During inference, we use $S{'}^{(1)}$ as the encoded trajectory embedding. The entire workflow of HiT-JEPA can be interpreted step-by-step as:
>     1. **Hierarchical trajectory abstraction creation**: we create higher-level trajectory abstractions $T^{(1)}$, $T^{(2)}$, and $T^{(3)}$ by a series of convolutions, max poolings, and layer normalizations according to Eq. (3)-(5). This process effectively aggregates raw point-level tokens into high-level semantic behavioral tokens. More details regarding the design rationale of this abstraction method can be referred to **response to reviewer 19oU (3/4)** and **response to reviewer uZkJ (2/3)**.
>     2. **Separate training at each JEPA level $l$**: at each level $l$, HiT-JEPA operates as a distinct predictive unit. we generate a set of embeddings:  target trajectory embedding $S^{(l)}$, context trajectory embedding $S{'}^{(l)}$, and predicted trajectory embedding $\widetilde{S}{'}^{(l)}$ from the context trajectory embedding $S{'}^{(l)}$ according to Eq. 6, Eq. 7, and Eq. 8. We also obtain a level-specific loss to ensure each JEPA layer is learning distinct, level-specific trajectory semantics.
>     3. **Hierarchical interactions**: to combine the multi-level trajectory semantics, we conduct a coarse-to-fine, high-to-low level gated fusion of attention weights according to Eq. (13)-(14). The justification of this fusion approach is that the level 1 encoders primarily focus on fine-grained physical dynamics, making them less robust against noisy signals. Consequently, by projecting and fusing attention weights derived from higher-level abstractions, we explicitly guide the fine-grained encoders with global context and comprehensively learn low-level trajectory dynamics and high-level movement behaviors.
>     4. **Inference**: to perform similarity search, we use the embedding $S{'}^{(1)}$ diretly coming out from the context encoder $E_\theta^{(l)}$ at level 1. Crucially, this embedding serves as a unified representation that integrates features from all three abstraction levels. Attention weights are fused top-down via hierarchical interaction to ensure that $S{'}^{(1)}$ encapsulates both high-level semantic intent and fine-grained kinematic details.
>
>
> 2. **To answer the question regarding the limited experiment of hierarchical representations**:
>     - we select similarity search as it is the **most fundamental and direct validation** for representation learning. The ability to preserve pairwise semantic proximity in the vector space is the theoretical prerequisite for any effective high-level application, such as clustering or behavioral analysis.
>     - we have visualized and explained in detail the **interpretability of hierarchical attention weights**, which is added in **Section 4.3** and **Appendices A.6** in our revised manuscript. We give a case study of the model attention at each JEPA level corresponding to a physical trajectory. We provide the raw attention map at each level (with each of the level 1 attention heads visualizations), the attention deviation heatmap, and attention statistical profiles to jointly analyze the trajectory semantics learned across all levels.
>     - **Distinct Novelty of the Framework**: HiT-JEPA proposes a novel architectural paradigm (Convolutional hierarchical Tokenizer + H3 cell embeddings + Hierarchical JEPA) designed to solve the unique challenges of diverse forms of urban trajectory data (GPS tracks and check-in sequences). Given competitive performance in urban trajectory similarity search and strong generalizations on zero-shot experimental settings, we believe this work constitutes a substantial contribution to the field of Trajectory Representation Learning.
>
> In summary, we believe the current experimental results on both quantitative similarity search and qualitative attention map visualization and case study suite rigorously validate HiT-JEPA as a novel and effective paradigm, therefore constitute a robust and self-contained contribution to the field.

---

> ### Author Response · Authors · 2025-12-03
> **Response to reviewer db7G (3/3)**
>
> ### **W2**: Validity of Evaluation Metrics and Discussion on Performance Limitations
>
> ### **A2**
>
> 1. **Acknowledgement to other tasks like trajectory clustering**.
> We thank the reviewer for suggesting other trajectory tasks, such as clustering. We fully acknowledge that these are critical and relevant downstream tasks for uncovering the learned representation quality in urban trajectory analysis. To reflect this, we have **expanded our related work (Section 2)** in our revision to briefly discuss the relevance and contribution of the clustering paper to the field. Due to the time constraints in the rebuttal phase, we do not make comprehensive comparisons on the clustering task. But we have visualized the K-Means clustering results by raw embeddings from HiT-JEPA in **Appendix A.9**. We demonstrate that even without a specific self-clustering design in recent clustering papers like [$\mathrm{E^2DTC}$](https://ieeexplore.ieee.org/stamp/stamp.jsp?tp=&arnumber=9458936) or [Yao *et al.* 2017](https://ieeexplore.ieee.org/stamp/stamp.jsp?tp=&arnumber=7966345), HiT-JEPA with solely regression loss on latent space can still generate visually discernible clusters.
>
> 2. **To answer the limited evaluation tasks**:
>     - **The validity of similarity search**: similarity is an intuitive, critical, and fundamental evaluation to assess the embedding quality, which is widely used in other fields such as embedding similarities for multi-modal data alignment, or text or document retrieval in NLP. Moreover, though the scope of the manuscript is about similarity search, we have thoroughly evaluated the **robustness and generalization** of HiT-JEPA. HiT-JEPA has demonstrated robustness against various downsampling and distortion rates in Table 1. The zero-shot experiments in TKY, NYC, and AIS(AU) in Table 1, and fine-tuning to approximate heuristic measures in Table 2 have demonstrated the strong generalizability of HiT-JEPA embeddings. These experiments prove that HiT-JEPA not only learns trajectory features in a fixed region but also more universal and general semantic behaviors underneath.
>     - **Masked encoding and decoding in the latent space**: JEPA-based methods differ from existing masked generative models, where the entire learning process is conducted in the latent space. The training objective prevents the model from memorizing the data itself, but the high-level semantics.
>     - **Visualizations and Interpretations**: We have visualized the HIT-JEPA embeddings by projecting them on the GPS coordinates in Section 4.2 and visualized the cross-level attention weights, followed by a case study in Section 4.3 to strengthen our novelty and effectiveness of our proposed hierarchical trajectory semantics structure.
>
> 3. **To answer the effectiveness and limitations of HiT-JEPA**:
>     - **Where effective**: HiT-JEPA excels in robustness in irregularly sampled trajectories such as T-drive, different forms of trajectories such as both GPS tracks and check-in sequences (TKY and NYC), and demonstrated strong cross-region generalization (zero-shot experiments in TKY, NYC, and AIS(AU)).
>     - **Where challenged**: HiT-JEPA rely on tokenized trajectory locations as the input, which poses a challenge of distinguishing extreme dense trajectory distribution in relative smaller cities like Porto compared to Beijing. However, we mitigate this by injecting various ratios of random masking at each JEPA level during pre-training. Experimental results show that the robustness is preserved by enabling the model to derive the latent dynamics and semantics from sole token inputs.
>     - **Architectural Constraint**: we have a brief discussion in the Appendices A.9 about the limit of our hierarchical interaction approach, which is only restrained for Transformer-based backbones as it operates on the attention weights.

---

### Author Response · Authors · 2025-12-03
**Summary of Revisions and Response to Area Chairs and Reviewers**

**We thank the Area Chairs and reviewers for your dedication to our research community and your constructive feedback. We have performed extensive rebuttal and revisions.**


We thank reviewer **db7G** for questions regarding the workflow and evidence of the hierarchical design, the limits of evaluation metrics, and the model's cross-region transferability.

We thank reviewer **uZkJ** for concerns regarding clarifications on trajectory-specific design, hierarchical interpretability, and CNN-based trajectory abstraction methods.

We thank reviewer **19oU** for requesting a response regarding additional baselines, hierarchical interpretability, and efficiency analysis.

We thank reviewer **ZHZ7** for raising questions regarding the novelty and the architectural choice of CNNs over Transformers.


## **Summary of our main contributions**

1. We propose HiT-JEPA, a novel Hierarchical Joint-Embedding Predictive Architecture for trajectory-specific similarity computation tasks. Different from existing hierarchical self-supervised learning methods, HiT-JEPA adopts a CNN-based trajectory abstraction method to bridge point-level trajectory local dynamics into higher-level semantic behaviors. By establishing such explicit trajectory semantic levels, HiT-JEPA learns level-specific trajectory semantics via independent JEPAs, then fuses them through a novel top-down hierarchical interaction mechanism, which ensures the learned representations are semantically comprehensive and structurally consistent.

2. Our analysis and visualizations reveal that HiT-JEPA learns distinct features across different levels of granularity, effectively capturing local pattern change and global trip intents.

3. We conduct extensive experiments on trajectory similarity search, demonstrating that HiT-JEPA can learn accurate representations that are robust against irregularly sampled trajectories, highly generalizable in zero-shot cross-city transfer, and computationally efficient.

## **Addressing main concerns from the reviewers**
We have responded to all reviewers' questions, and we summarize the main concerns and how we responded below:

### **Methodology & Novelty**
- **The workflow and evidence of the hierarchical design.** We have explained in detail the workflow of HiT-JEPA in our **Response to reviewer db7G (2/3)**, referring to the method descriptions in our manuscript.
- **Trajectory-specific design.** We have thoroughly explained our justification of each main component of our hierarchical structure design and provided experimental evidence to prove the effectiveness. We changed the keywords in lines `086-087` to concretize our key design rationale.
- **Rationale of CNN-based trajectory abstraction.** We have explained our justifications of using CNNs as the trajectory abstraction method in **Response to reviewer ZHZ7 (3/4)** and **Response to reviewer uZkJ (2/3) (Part 1 of 2)**.

### **Experiments & Evaluations**
- **Interpretability of HiT-JEPA.** We have provided a case study with attention weight visualizations at each JEPA level corresponding to a physical trajectory. Please see our `revision in Section 4.3 and Appendix A.6`.
- **More baseline comparisons.** We have compared with one more baseline method, please see `revision in Table 1 and Table 2`.
- **Limits of similarity task.** We have restated that the similarity search is the fundamental and direct validation in **Response to reviewer db7G (3/3)**. Moreover, we have also conducted a simple clustering experiment and visualized the clustering results in `Appendix A.9`, which shows that HiT-JEPA can generate visually discernible clusters even without a clustering-specific design.
- **Additional overall performance boost with hyperparameter change.** We have updated the equations `Eq. (3)-(5) and lines 209-212`, as well as our experimental results for HiT-JEPA. Please see `revisions for all tables and Figs 6 and 7`.

### **Efficiency**
We provide training time comparisons between HiT-JEPA and all baselines, please see `revision in Appendix A.7`.

### **Other Revision**
We have also updated an inaccurate description at `Eq. 14`. And due to the "checkerboard artifact" from the transpose convolution, we have changed it to bilinear interpolation in `Eq. 13`.


**We sincerely thank all ACs and reviewers for their time in helping us improve HiT-JEPA. We believe that the additional experiments and clarifications strengthen the quality and rigor of our manuscript.**

---

### Meta-Review · Area_Chair_T44c · 2025-12-29

**Summary:**

The main concerns focused on novelty and scope. Several reviewers felt the method builds heavily on existing JEPA and hierarchical SSL ideas, with limited conceptual departure beyond adaptation to trajectories. There were also questions about evaluation breadth, as experiments center almost exclusively on similarity search, and whether hierarchical semantics are truly validated beyond retrieval metrics. Efficiency and interpretability were raised but seen as secondary.

**Reviewer Concerns:**

The rebuttal did a solid job addressing clarity and empirical gaps. It clarified the hierarchical workflow, added attention visualizations, expanded baseline comparisons, provided efficiency analysis, and justified CNN-based tokenization and H3 grids with both reasoning and experiments. These responses strengthened the paper technically. However, core novelty concerns remain partially outstanding. While the authors better articulated trajectory-specific design choices, some reviewers may still view the contribution as an incremental extension of existing hierarchical JEPA/HSSL paradigms, with limited exploration beyond similarity search to fully justify the hierarchy’s necessity.

**Reviewer Scores:**

Reviewer db7G: Likely a small upward shift, as workflow clarity, transferability, and interpretability concerns were addressed.

Reviewer 19oU: Possibly unchanged or slightly lower

Reviewer uZkJ: Might move slightly higher, because many technical questions were answered, but novelty concerns likely persist.

Reviewer ZHZ7: Remaining borderline due to novelty concern.

---

### Decision · Program_Chairs · 2026-01-26

Reject